# Quality by Design and In Silico Approach in SNEDDS Development: A Comprehensive Formulation Framework

**DOI:** 10.3390/pharmaceutics17060701

**Published:** 2025-05-27

**Authors:** Sani Ega Priani, Taufik Muhammad Fakih, Gofarana Wilar, Anis Yohana Chaerunisaa, Iyan Sopyan

**Affiliations:** 1Doctoral Program of Pharmacy, Faculty of Pharmacy, Universitas Padjadjaran, Sumedang 45363, Indonesia; sani24002@mail.unpad.ac.id; 2Faculty of Mathematics and Natural Sciences, Bandung Islamic University, Bandung 40116, Indonesia; taufikmuhammadf@unisba.ac.id; 3Department of Pharmacology and Clinical Pharmacy, Faculty of Pharmacy, Universitas Padjadjaran, Sumedang 45363, Indonesia; g.willar@unpad.ac.id; 4Department of Pharmaceutics and Technology of Pharmacy, Faculty of Pharmacy, Universitas Padjadjaran, Sumedang 45363, Indonesia; anis.yohana.chaerunisaa@unpad.ac.id

**Keywords:** SNEDDS, optimization, quality by design, design of experiment, molecular modelling

## Abstract

**Background/Objectives**: The Self-Nanoemulsifying Drug Delivery System (SNEDDS) has been widely applied in oral drug delivery, particularly for poorly water-soluble compounds. The successful development of SNEDDS largely depends on the precise composition of its components. This narrative review provides an in-depth analysis of Quality by Design (QbD), Design of Experiment (DoE), and in silico approach applications in SNEDDS development. **Methods**: The review is based on publications from 2020 to 2025, sourced from reputable scientific databases (Pubmed, Science direct, Taylor and francis, and Scopus). **Results**: Quality by Design (QbD) is a systematic and scientific approach that enhances product quality while ensuring the robustness and reproducibility of SNEDDS, as outlined in the Quality Target Product Profile (QTPP). DoE was integrated into the QbD framework to systematically evaluate the effects of predefined factors, particularly Critical Material Attributes (CMAs) and Critical Process Parameters (CPP_S_), on the desired responses (Critical Quality Attributes/CQA), ultimately leading to the identification of the optimal SNEDDS formulation. Various DoEs, including the mixture design, response surface methodology, and factorial design, have been widely applied to SNEDDS formulations. The experimental design facilitates the analysis of the relationship between CQA and CMA/CPP, enabling the identification of optimized formulations with enhanced biopharmaceutical, pharmacokinetic, and pharmacodynamic profiles. As an essential addition to this review, in silico approach emerges as a valuable tool in the development of SNEDDS, offering deep insights into self-assembly dynamics, molecular interactions, and emulsification behaviour. By integrating molecular simulations with machine learning, this approach enables rational and efficient optimization. **Conclusions**: The integration of QbD, DoE, and in silico approaches holds significant potential in the development of SNEDDS. These strategies enable a more efficient, rational, and predictive formulation process.

## 1. Introduction

The pharmaceutical industry faces challenges in drug formulation and development, particularly for BCS classes II and IV compounds [1]. Approximately 40% of active pharmaceutical ingredients fall into these two BCS categories [2,3]. They exhibit low water solubility, poor gastrointestinal dissolution, and low bioavailability. Strategies to enhance dissolution and bioavailability are critical aspects of drug formulation [3]. Various strategies have been explored to enhance the solubility of poorly water-soluble drugs, facilitate their dissolution, optimize absorption, and ultimately lead to improve their therapeutic efficacy [4]. The Self-Nanoemulsifying Drug Delivery System (SNEDDS) is a dosage form that markedly enhances the dissolution rate of active pharmaceutical ingredients, particularly for oral administration [5]. SNEDDS is a pre-concentrate nanoemulsion consisting of oil, surfactants, and co-surfactants that spontaneously form a nanoemulsion system with gentle agitation in the gastrointestinal tract [6]. The small size of the oil globules increases the contact area of the active ingredient with the gastrointestinal fluids, thereby enhancing its dissolution. Additionally, the presence of surfactant/co-surfactant components, drug absorption through the lymphatic system, inhibition of P-glycoprotein (P-gp) efflux, and prevention of intraenterocyte metabolism are other mechanisms that contribute to the improved bioavailability of drugs in SNEDDS [7,8].

The development of SNEDDS presents several formulation challenges owing to the complex interrelationships among its key components (oil, surfactant, and co-surfactant), which significantly influence system stability, emulsification efficiency, and drug-loading capacity [8]. Achieving the optimal composition is essential to ensure the formation of a stable nanoemulsion with optimal properties [6]. To overcome these issues, quality by design (QbD) provides a structured approach for SNEDDS development [9]. In QbD, product quality is ensured by comprehensively understanding and controlling formulation and manufacturing variables to achieve consistency and robustness [10,11]. In QbD, identifying Critical Material Attributes (CMAs) and Critical Process Parameters (CPPs) is crucial for maintaining Critical Quality Attributes (CQAs) and ensuring a systematic approach to pharmaceutical development. Guidelines such as ICH Q8(R2), Q9, and Q10 require the adoption of QbD in pharmaceutical development, which addresses pharmaceutical development, quality risk management, and pharmaceutical quality systems [12,13,14,15,16].

Design of Experiments (DoE) plays a crucial role in QbD by enabling a systematic and efficient approach to optimize pharmaceutical formulations and manufacturing processes. DoE allows for the simultaneous evaluation of multiple factors, identification of their interactions, and assessment of their impact on CQAs, thereby providing a comprehensive understanding of the formulation (CMAs) and process variables (CPPs) [13]. Design of Experiments (DoE) has been widely adopted in the development of SNEDDS and is well recognized for its effectiveness in identifying critical formulation factors and ensuring the production of high-quality, consistent formulations [17,18]. Integrating QbD and DoE in SNEDDS formulation significantly enhanced time and cost efficiency. QbD promotes a structured approach that minimizes trial and error, resulting in more predictable outcomes and reducing the need for frequent revisions during development.

In addition to QbD, recent advances have highlighted the potential of in silico approaches as complementary tools for optimizing SNEDDS formulations. In silico methods, such as molecular dynamics simulations, provide detailed insights into drug–excipient interactions and physicochemical behaviours at the molecular level. These computational techniques can predict miscibility and stability, thereby reducing time and cost [19]. By integrating an in silico approach within the QbD framework, researchers can enhance the rational design and understanding of SNEDDS formulations, ultimately improving formulation performance and stability.

QbD and DoE have been widely applied to SNEDDS development; however, no comprehensive review has specifically examined their implementation in this field. This narrative review provides an in-depth analysis of QbD and DoE applications in SNEDDS formulation development, drawing on recent studies published between 2020 and 2025. It presents a structured, step-by-step examination and discussion of outcomes. Additionally, this review highlights the integration of an in silico approach in SNEDDS development and discusses future perspectives to guide ongoing and upcoming research in the field.

## 2. Basic Principles of SNEDDS Formulation

SNEDDS is a drug delivery system of oil, surfactants, and co-surfactants that spontaneously forms a nanoemulsion in water with gentle agitation (Figure 1) [20]. Unlike nanoemulsions or microemulsions, SNEDDS exists as an anhydrous preconcentrate and only forms a nanoemulsion in situ upon contact with gastrointestinal fluids. This unique characteristic contributes to enhanced physical stability during storage, as there is no aqueous phase present that could lead to phase separation or degradation before administration. Spontaneous emulsification of SNEDDS occurs because the free energy (ΔG) approaches zero, achieved by the flexible interface of the surfactant and co-surfactant molecules, which reduces surface tension [21]. Compared to other lipid-based systems like nanostructured lipid carriers (NLCs), solid lipid nanoparticles (SLNs), and liposomes, SNEDDS offers the advantages of ease of production and avoids aggregation risks as it is a preconcentrated nanoemulsion that only forms a nanosystem in the gastrointestinal tract [5,22].

The SNEDDS consists of three main components: oil, surfactant, and co-surfactant [23]. Oil serves as the primary carrier for drugs, enhancing the solubility of the active ingredients in the lipid phase. The oil used can be natural or synthetic and can be selected based on its ability to dissolve the active ingredient [24,25]. Natural oils are commonly chosen based on their safety and biocompatibility considerations [5]. Some studies have used natural oils with pharmacological activities that align with the active ingredient to form a bioactive SNEDDS formulation [26,27,28]. Many SNEDDS formulations utilize synthetic oils primarily because of their consistent composition, higher solubilization capacity, and improved stability compared to natural oils [5,29,30]. Surfactants reduce interfacial tension, facilitating the formation of uniformly sized nanoemulsion globules [5]. Non-ionic surfactants are commonly used because of their low toxicity and ability to stabilize emulsions across a wide pH range [31]. Beyond emulsification, non-ionic surfactants like Tween 80 and Cremophor EL/RH40 enhance membrane fluidity and inhibit efflux transporters, improving drug bioavailability [32]. Co-surfactants enhance nanoemulsion stability and homogeneity by improving interfacial fluidity while simultaneously reducing the required concentration of surfactants, which helps minimize potential toxicity risks. Common co-surfactants include short-chain alcohols and glycols, such as propylene glycol, ethanol, glycerol, polyethylene glycol, and Transcutol [5,33,34].

As previously discussed, the selection of oils, surfactants, and co-surfactants in SNEDDS is primarily based on their ability to solubilize the drug and promote spontaneous emulsification. However, their safety profile, particularly for oral administration, is equally important. Since SNEDDS typically contain relatively high levels of excipients, it is essential to use components that either have Generally Recognized as Safe (GRAS) status or are listed in the FDA Inactive Ingredient Database for oral use. Many commonly used SNEDDS excipients meet these criteria, including medium-chain glycerides, oleic acid, castor oil, Polysorbate 80, Polysorbate 20, lecithin, Cremophor EL, propylene glycol, Transcutol P, and polyethylene glycol 400 (PEG 400), which the FDA accepts for oral use within specified concentration limits. Both the type and concentration of excipients must be carefully considered. These safety considerations are incorporated into the design of experiments and are fundamental to the Quality by Design framework, ensuring that formulation optimization does not compromise patient safety [35,36].

SNEDDS development generally involves several stages. Each stage is crucial for selecting proper excipients and achieving efficient self-nanoemulsification.

### 2.1. Solubility Testing

The first stage is solubility testing, which aims to determine the solubility of the active pharmaceutical ingredient (API) in various oils, surfactants, and co-surfactants. Because SNEDDS relies on dissolving the drug in lipid-based excipients, selecting components with a high solubilizing capacity is essential. Oils improve drug absorption, whereas surfactants and co-surfactants facilitate emulsification [5,37]. This step ensures that the final formulation can carry an adequate drug load without precipitation by identifying excipients that can dissolve the maximum amount of API [5,37].

### 2.2. Emulsification Efficiency

Emulsification efficiency, also referred to as emulsification capability, evaluates the ability of a given combination of oil, surfactant, and co-surfactant to spontaneously form a nanoemulsion upon contact with an aqueous medium [38,39,40,41]. Because the oil phase is generally selected based on the solubility results from the previous stage, this test was primarily conducted to determine the most effective combination of surfactant and co-surfactant. Rapid and spontaneous emulsification is a key characteristic of SNEDDS, allowing the drug to be efficiently dispersed in the gastrointestinal tract. This procedure involved introducing a small volume of the preconcentrate (SNEDDS formulation) into an aqueous medium under mild stirring to simulate the gastrointestinal environment. The emulsification efficiency was assessed visually, and the globule size and/or % transmittance were measured using an appropriate instrument [39,42].

### 2.3. Construction Diagram Pseudo-Ternary

Once suitable excipients are selected, a pseudo-ternary phase diagram defines the range of oil, surfactant, and co-surfactant concentrations that lead to stable nanoemulsion formation. The optimum ratio of surfactant to co-surfactant can also be determined by observing the area of the nanoemulsion formation within the phase diagram. Because SNEDDS formulations require a careful balance between these components, mapping the phase diagram helps to effectively visualise the regions where nanoemulsification occurs [34,43,44]. Oil, surfactant, and co-surfactant mixtures were prepared in various ratios and titrated with water while stirring. Each system was observed, and the results were plotted on a ternary diagram to define the nanoemulsion region [45]. Evaluation of self-emulsification and turbidity after dilution is an alternative to titration methods for assessing the effectiveness of SNEDDS formation [46,47].

### 2.4. Optimization of SNEDDS Formulation

After obtaining the SNEDDS formation area through the analysis of the pseudo-ternary phase diagram, the next step was to optimize the SNEDDS formulation. This optimization process can be performed by creating several formula variations. Variations are generally applied to the oil-to-Smix ratio and surfactant-to-cosurfactant ratio to select the formulation that produces the optimum SNEDDS characteristics [43]. SNEDDS was prepared by mixing oil, surfactant, and co-surfactant in specific proportions, followed by homogenization using magnetic stirring or ultrasonication methods until a stable mixture was formed. Several approaches can be used to determine an optimal formulation. Beyond these conventional approaches, SNEDDS optimization can be further enhanced by implementing a QbD strategy, enabling a more structured and risk-based development process.

## 3. The Principle of QbD in Product Development

ICH Q8, known as “Pharmaceutical Development”, offers a structured framework to support pharmaceutical development and ensure drug product quality, efficacy, and safety. These guidelines describe a systematic approach to drug development, emphasizing the application of scientific principles and risk-based assessments to achieve a well-defined manufacturing process. It encourages a transition from conventional empirical methods to a more structured and knowledge-driven approach closely aligned with the core concepts of QbD [16,48]. Drug development based on QbD consists of several phases (Figure 2).

The development of pharmaceutical products following the QbD approach begins by defining the Quality Target Product Profile (QTPP), which outlines the desired characteristics of the final product, including dosage form, route of administration, strength, and stability. This is a strategic reference point for the formulation, development, and process design [16,49]. Once the QTPP is established, the next step is to identify Critical Quality Attributes (CQA). CQAs are essential properties that must be controlled to ensure the product quality, safety, and efficacy. These CQAs include physical, chemical, biological, and microbiological attributes, and their identification requires an understanding of the influence of raw materials and process conditions on product performance [16,49,50]. Risk assessment systematically links the formulation and process variables to product quality. It evaluates the relationship between Critical Material Attributes (CMAs) and Critical Process Parameters (CPPs) for CQAs. Ishikawa diagrams and Failure Mode and Effects Analysis (FMEA) are often employed to assess and prioritize potential risks, ensuring that formulation and process development address factors with the highest potential to impact quality [16,51,52].

Based on these risk assessments and experimental data, a design space was established to define the acceptable range of formulation and process parameters, ensuring consistent product quality. This is typically achieved using statistical modelling approaches, such as the Design of Experiments (DoE), which systematically evaluate how variable factors influence CQAs. The design space is a roadmap for the development of robust formulations and processes [10,16,53]. A control strategy consisting of raw material specifications, process monitoring, and final product testing is implemented to ensure long-term product quality. Finally, lifecycle management ensures continuous improvement through ongoing monitoring, real-time data collection, and process verification using tools like Process Analytical Technology (PAT). Any post-approval changes are guided by prior knowledge and risk-based approaches, ensuring efficiency and regulatory compliance throughout the product life cycle [14,16].

## 4. Application of QbD in SNEDDS Development

The conventional approach to developing SNEDDS commonly only allows the observation of One Factor at a Time (OFAT), making it less efficient in understanding the interactions between formulation variables. This limitation has led to the implementation of a more systematic and scientific approach, known as QbD. Through QbD, SNEDDS development not only focuses on formulation optimization but also enables a deeper understanding of the influence of each factor on product quality (CQAs). Consequently, the formulated product exhibits a better controlled quality and consistency than the conventional empirical approach [54,55]. Several studies have developed SNEDDS formulations based on the QbD approach following the general stages outlined earlier (Table 1) [38,56,57,58,59,60,61,62]. The process began by determining the QTPP, CQAs, CMAs, and CPPs, followed by a risk assessment to evaluate the impact of CMAs and CPPs on CQAs. Finally, a design space was established. However, the number of studies implementing a comprehensive QbD approach is relatively limited. Most studies have focused on determining the CQAs, CPPs, and CMAs and designing experiments to optimize the formulation. In many cases, QTPP and risk assessment have not yet been clearly defined [63].

### 4.1. Define of QTPP in SNEDDS Development

The QTPP is the expected final quality specification of a product. In pharmaceutical dosage forms, such as SNEDDS, QTPP serves as a guideline to ensure that the formulation meets the optimal quality, safety, and efficacy criteria. The QTPP, one of the key aspects of QbD, serves as the basis for identifying CQAs [64]. As shown in Table 1, several elements of the QTPP in the SNEDDS dosage form include clinical target, dosage form, dosage type, dosage strength, route of administration, packaging/container closure system, pharmacokinetic parameters, stability, and alternative administration methods. Specific targets were established for each QTPP element, along with justifications explaining the rationale behind their selection. These targets ensure that the formulation satisfies the required quality, efficacy, and safety standards [65,66]. Table 2 shows the common QTPP elements, their targets, and justifications for SNEDDS formulation.

### 4.2. Identify CQAs in SNEDDS Development

The process begins by defining the quality attributes of the product and collectively contributing to meeting the predefined QTPP. Once these quality attributes are identified, each attribute is further analyzed to assess its potential impact on product performance, allowing for the identification of Critical Quality Attributes. An attribute is critical if it significantly affects the safety, efficacy, or stability of a product. This is critical when variations substantially change the quality of the final product [63,64,67]. In the context of SNEDDS, several common attributes that may be categorized as CQAs include the globule size, emulsification time, % transmittance, drug release/dissolution, zeta potential, and polydispersity index (Table 3). These attributes directly affect the performance of SNEDDS, particularly in enhancing the bioavailability of poorly soluble drugs. Similar to establishing the QTPP, strong justification must also support the identification of the CQA. This justification should include a scientific rationale and experimental data to validate the selection of specific CQA. Generally, not all CQAs are implemented; only those most suitable for particular research conditions are selected [59,68].

### 4.3. Risk Assessment in SNEDDS Development

In the QbD approach, risk assessment is used to identify factors that may affect the quality of pharmaceutical products. After the CQAs were established in the previous stage, a risk assessment was conducted to evaluate factors that may cause variations in product quality, including CMAs and CPPs. CMAs refer to the inherent characteristics and attributes of raw materials, whereas CPPs refer to manufacturing process parameters that influence the quality of the final product. CMAs and CPPs have a significant impact on CQAs and are controlled through risk assessments to ensure quality consistency and optimal product performance. Risk assessment typically begins with identifying potential risks using tools such as the Ishikawa diagram, followed by assessing the impact of each identified risk using appropriate methods. According to ICH Q9’s quality risk management section, the risk management process can be conducted using various approaches such as Failure Mode and Effects Analysis (FMEA) and Risk Ranking and Filtering (REM). The selection of an appropriate method depends on the level of risk, process complexity, and data availability. REM is commonly applied for initial screening, followed by FMEA when a more detailed analysis is required, particularly for high-risk processes. Although REM does not necessarily have to precede FMEA, combining both methods is often considered the best practice to achieve a comprehensive risk assessment [15,69].

In SNEDDS development, nearly all researchers used the Ishikawa diagram for risk identification (Table 1). The Ishikawa diagram, also known as the fish-bone diagram, is used to systematically identify and analyze the factors contributing to a problem or risk within a process. It is widely applied in quality control and risk management, including in pharmaceutical formulations, based on the QbD approach. The diagram resembles a fish skeleton, where the main issue (effect) is positioned at the head of the fish, while the potential causes are categorized into branches along the fishbone. To construct an Ishikawa diagram, the first step is to define the main problem, identify the primary cause categories, add specific contributing factors to each category, and then conduct an in-depth analysis to determine the factors with the most significant impact. These causes are typically grouped into six categories: man (human), machines (equipment), materials (raw materials), methods (processes), measurements (evaluations), and the environment (external factors) [70].

Several studies on SNEDDS development using the QbD approach have demonstrated various strategies for conducting risk assessments. Some articles only described the construction of the Ishikawa fish-bone diagram without further elaborating on the quantitative risk assessment methods. In contrast, others describe subsequent stages using Risk Ranking and Filtering (REM), a Risk Assessment Matrix (RAM), or FMEA to classify and prioritize risks systematically. REM ranks risks by assigning scores to predefined criteria, typically severity and probability, and then sorts risks from highest to lowest priority. RAM evaluates risks based on a matrix that cross-references the severity and likelihood to classify them as low-, medium-, or high-risk. FMEA, a more comprehensive method, calculates an RPN by multiplying the scores for severity (S), probability (P), and detectability (D), providing a quantitative basis for prioritization and control decisions. A higher RPN value indicates a higher risk level that requires appropriate control measures. The QbD-based SNEDDS formulation of Cinacalcet HCl was developed using the FMEA method. The results showed that high RPN values (score >250) were observed for the oil concentration, surfactant concentration, co-surfactant concentration, absorbent concentration (in S-SNEDDS preparation), stirring speed, stirring time, and temperature [58,69].

The risk analysis results for SNEDDS generally indicated that the concentrations of oil, surfactant, and co-surfactant are high-risk factors that must be carefully controlled in SNEDDS formulations. Therefore, most SNEDDS formulations designate these components as CMAs individually (oil concentration, surfactant concentration, and co-surfactant concentration) or as a combined Smix concentration (surfactant + co-surfactant). The application of process-related high-risk factors (CPPs) in SNEDDS formulations is limited. This is because SNEDDS is a nanoemulsion pre-concentrate designed to spontaneously form nanoemulsions upon contact with gastrointestinal fluids, unlike conventional emulsions or other nanoformulations that require high-pressure homogenization. The simplicity of the SNEDDS formation implies that only a few process parameters pose significant risks. Furthermore, in SNEDDS formulations, CMAs, such as the type and concentration of oil, surfactant, and co-surfactant, have a more substantial impact on product stability and performance than the process parameters. Consequently, CMAs have received more attention than CPPs in SNEDDS formulations. Although relatively rare, some studies of SNEDDS have implemented CPPs. For example, sonication time has been identified as a CPP in the SNEDDS formulations of docetaxel and bedaquiline. Increasing the sonication time enhanced the sonication energy and mechanical force, which ultimately broke the oil globules into smaller globules, thereby significantly reducing their size [57,59]. Figure 3 illustrates the relationship between CQA, CPP, and CMA in the development of the SNEDDS.

The discussion on QTPP, CQA, CMA, and CPP has predominantly focused on the development of SNEDDS for oral administration. This is mainly due to the extensive research and well-established applications of SNEDDS in enhancing the solubility and bioavailability of poorly water-soluble drugs via the oral route. However, some studies have also investigated the use of SNEDDS for non-oral routes such as ocular, intranasal, and transdermal administration, although the number of such studies remains relatively limited [71,72,73,74]. For these alternative routes, adjustments in QTPP, CQA, CMA, and CPP are necessary to accommodate the distinct physiological conditions, absorption mechanisms, and therapeutic objectives, despite many aspects being shared across different routes. For example, in one ocular SNEDDS development study using DoE, the CMAs included variations in oil, surfactant, and co-surfactant concentrations, as well as surfactant type. The CQAs were defined based on parameters such as globule size, emulsification time, and % transmittance after dilution. Although CMAs and CQAs were similar to those considered in oral SNEDDS development, key differences appeared in the process parameters and testing conditions. For instance, dilution was conducted using simulated tear fluid (STF, pH 7.4) at a 1:10 volume ratio under gentle stirring (30 rpm) at 35 °C to closely mimic the ocular environment, including corneal surface conditions and eyelid movement [72].

## 5. Application of DoE in SNEDDS Development

In QbD, the Design of Experiments (DoE) is a systematic approach for understanding the relationship between CMAs, CPPs, and CQAs. Compared to other optimization methods, such as OFAT, DoE has the advantages of studying multiple factors simultaneously, providing more accurate information and identifying interactions between factors affecting product quality. This approach is expected to reduce development costs and time while improving the reproducibility of the results [75,76]. DoE in pharmaceutical product development can be carried out using various types of design, including mixture design (simplex lattice design, extreme vertices design, and optimal mixture design), response surface methodology (Box–Behnken design and Central Composite Design), and factorial design. Various experimental design methods can be applied as optimization tools in the development of SNEDDS formulations, each offering distinct advantages [77,78,79].

### 5.1. Mixture Design

Mixture design has been widely applied in the development of SNEDDS formulations [39,40,41,72,80,81,82,83,84,85,86,87,88,89,90,91,92,93,94,95,96]. Mixture design is a specialized approach within the DoE framework utilized to optimize pharmaceutical formulations. This method is particularly relevant when the response of the system is determined by the relative proportions of the multiple components in a formulation. A fundamental characteristic of mixture design is that the total sum of all elements in the formulation remains constant (100% or one) [97,98]. Composition adjustments were made by modifying the relative ratios of the components, without altering the overall quantity of the mixture. This makes the mixture design well-suited for developing SNEDDS formulations, mainly when the defined factors are the oil, surfactant, and co-surfactant concentrations. The data in Table 4 show that the mixture design can yield a final SNEDDS formulation with a 100% proportion of oil, surfactant, and co-surfactant. However, the table also highlights that applying mixture design is limited to evaluating CMAs because it does not consider CPPs as contributing factors in the formulation process. If the process parameters are not considered critical under certain conditions, mixture design is an appropriate choice for optimizing SNEDDS formulations.

Mixture design is classified into several types based on the structure and distribution of the experimental points or the number of components in the mixture. Data showed that extreme vertex mixture design, simplex lattice design, and optimal mixture design have been utilized to optimize SNEDDS formulations, with the optimal mixture design being the most frequently used. An optimal mixture design is particularly valuable when dealing with formulations that impose specific constraints on the component concentrations. It strategically identifies experimental points to maximize statistical efficiency, making it an excellent choice when resources and time are limited. This approach helps reduce the number of required experiments while still ensuring the development of reliable predictive models. There are various types of optimal designs, including D-optimal and I-optimal. The I-optimal design was utilized for the development of ibrutinib-loaded SNEDDS [90]. The D-optimal design was applied to the SNEDDS of alpha-lipoic acid, candesartan, cefixime, exenatide, rosuvastatin, voxelotor, and a combination of ezetimibe–atorvastatin. This is one of the most commonly applied DoE in developing SNEDDS [40]. The main difference between D-optimal and I-optimal designs lies in their optimization criteria. D-optimal design focuses on maximizing the determinant of the information matrix to ensure precise estimation of model coefficients, while I-optimal design minimizes the average prediction variance across the design space.

Another mixture design used in SNEDDS development is the simplex lattice design (SLD). SLD is a statistical method for analyzing multiple variables and their interactions and identifying relationships between factors and responses to determine the optimal combination. The simplex lattice design systematically arranges the experimental points across the formulation space in a structured, grid-like pattern [99]. SLD has been used to develop SNEDDS containing valsartan, potassium diclofenac, and a combination of neomycin sulfate and thioctic acid [82,83,84]. Another type of mixture design applied to SNEDDS is the extreme vertex design, which has been used in the SNEDDS of glimepiride-rosuvastatin and febuxostat. Theoretically, an extreme-vertex mixture design distributes points evenly or is based on statistical optimization, which concentrates on testing formulations at the extreme boundaries of the allowable composition space [100,101].

### 5.2. Response Surface Methodology

Response surface methodology (RSM) is another design widely applied in SNEDDS development [9,38,47,57,58,59,60,102,103,104,105,106,107,108,109,110]. RSM is a statistical technique widely used for optimizing processes by evaluating the relationships between multiple independent and response variables. This approach is beneficial in experimental design as it reduces the required trials while providing a comprehensive understanding of factor interactions. RSM employs mathematical models to describe the response surface and predict the optimal conditions [111,112]. The Box–Behnken Design (BBD) and Central Composite Design (CCD) are the most commonly used RSM designs. BBD is a three-level design that efficiently models factor interactions while avoiding extreme (corner) points in the experimental space. By contrast, CCD consists of factorial, axial, and central points, making it a more flexible design that explores a broader experimental space, mainly due to the inclusion of axial points [113].

Unlike the mixture design, the total proportion of the mixture components (oil, surfactant, and co-surfactant) cannot be directly fixed at 100% in the RSM, as shown in Table 5. This is because selecting experimental points in the RSM focuses more on exploring the effects of individual variables than on maintaining a fixed total composition [111]. Therefore, further calculations are required to determine the optimal percentage formulation. Several studies of SNEDDS using RSM have employed volume or weight units for component factors as alternatives to percentage-based compositions [58,59]. Apart from enabling efficient point selection and experimental design, this method allows one to assess the influence of the manufacturing process (CPPs). This aspect is not feasible in mixture design, as observed in studies of SNEDDS for Bedaquiline [57], Docetaxel [59], and Venetoclax [110]. Additionally, the CMAs analyzed in the RSM were not limited to the mixture proportions. They can also include other variables, such as the surfactant/co-surfactant ratio, dosage, and type of surfactant used [104,108]. The RSM allows for including numerical and categorical variables, whereas the mixture design is restricted to numerical variables [114].

### 5.3. Factorial Design

Several studies have employed factorial design in addition to mixture design and RSM in the development of SNEDDS formulations [77,116,117,118,119,120,121,122,123,124]. Factorial design is a statistical approach used to evaluate the effects of multiple independent factors and their interactions on the responses. It includes a Full Factorial Design, which examines all possible combinations of factor levels, and a fractional factorial design, which reduces the number of runs by selecting a subset of combinations. The factorial design incorporates both numerical and categorical factors [77]. The selection of experimental points represents a key distinction between factorial design and RSM. In a factorial design, experimental points are typically placed at low and high levels, with an additional intermediate level in three-level designs. This approach efficiently identifies main effects and interactions but is less suitable for capturing nonlinear responses. In contrast, RSM enables the modelling of response surface curvature and higher-order relationships. In SNEDDS development, factorial design is commonly used for screening critical formulation and process parameters (CMAs and CPPs), while RSM is preferred for optimization [116].

Factorial design, mainly Full Factorial Design, requires many experimental runs, owing to its approach of testing all possible factor combinations. Many experiments can be impractical, particularly in SNEDDS optimization, where multiple formulation variables must be considered. To address this issue, some researchers have minimized the number of runs by reducing the number of factors studied. For instance, instead of treating oil, surfactant, and co-surfactant concentrations as separate CMAs, researchers may combine surfactant and co-surfactant into a single Smix factor and optimize the oil concentration and Smix concentration (Table 6). This approach reduces the number of factors and may also reduce the experimental burden. However, alternative designs, such as fractional factorial or response surface methods, are often preferred for further efficiency while preserving the key formulation interactions [118,120].

## 6. Optimization Stages in SNEDDS Development Based on DoE

DoE is applied in SNEDDS optimization to achieve two main objectives: (1) analyze the influence of CMAs and CPPs on CQAs and (2) obtain an optimal formulation that meets the desired CQA criteria. Several systematic steps must be followed in order to achieve these goals.

### 6.1. Determination of Factors and Responses

In DoE, the terms, factors and responses, play a crucial role. A factor or independent variable refers to a variable that can influence the outcome of an experiment and is classified as a CMA or CPP [68,126]. By contrast, a response or dependent variable represents the measured outcome of the experiment, corresponding to the CQA. As explained in the QbD section, determining CQAs is the first step and must be supported by proper justification [56,57]. Based on Table 4, Table 5 and Table 6, the responses of the SNEDDS formulation included the globule size, emulsification time, polydispersity index (PDI), % transmittance, % drug release, zeta potential, and drug loading. However, researchers are not required to designate all these parameters as responses. Instead, they should be selected with proper justification, focusing on the quality attributes that are most relevant to the desired product performance. The most commonly established responses are the globule size, emulsification time, and drug release. Globule diameter is a critical parameter to ensure that SNEDDS spontaneously forms a nanoemulsion upon contact with aqueous fluids, typically measuring less than 200 nm. The emulsification time reflects the ability of the system to create a nanoemulsion spontaneously upon contact with an aqueous fluid. Meanwhile, the percentage of drug release is significant when the primary objective of SNEDDS development is to enhance the dissolution of poorly water-soluble drugs [5].

After establishing the responses, a risk assessment was conducted to identify factors significantly affecting product quality [56]. The data in the tables indicate several commonly used factors in SNEDDS formulation. The most frequently identified factors were oil, surfactant, and co-surfactant concentrations. Some studies combine the influence of surfactant and co-surfactant into a single factor, the Smix concentration [104,116]. This approach helps to reduce the number of factors, minimize the number of experimental runs, and allow for additional factors, such as the oil/surfactant/co-surfactant type [108,125].

The tables show that only a limited number of studies included CPP as a factor in their experimental designs. The CPP-related factors investigated in SNEDDS development included sonication time, stirring speed, and stirring time [57,59,110]. Other studies that do not include CPP as a factor consider that the composition of SNEDDS components significantly affects product quality more than the manufacturing process [81,83].

### 6.2. Selecting Experimental Design

Selecting the most appropriate DoE approach involved basing it on study objectives and characteristics. However, the available publications generally do not explain the reasons for choosing a particular DoE type in detail. As previously discussed, three main DoE approaches can be applied in SNEDDS optimization: mixture design, response surface methodology, and factorial design. Factorial design is commonly used to screen for significant factors, helping identify key variables that influence SNEDDS performance. Response surface methodology and mixture design are preferred for optimization, mainly when dealing with nonlinear relationships. When the optimization focuses solely on the composition of the SNEDDS components, oil, surfactant, and co-surfactant, mixture design (e.g., optimal mixture design) is ideal. This method ensures that the proportion of these three components is 100%, thereby providing an optimal formulation based on their relative ratios. However, if optimization involves additional factors beyond composition, such as numerical or categorical variables and Critical Process Parameters (CPPs), RSM becomes more suitable. RSM optimizes the formulation composition (CMAs) and process parameters (CPPs) while effectively capturing the interaction effects and nonlinear relationships that influence the SNEDDS characteristics [77,97,113,127].

### 6.3. Establishment of Experimental Points

The experimental points can be established after defining the factors/responses and selecting the DoE type. This process ensured the experimental design was efficiently structured, allowing systematic data collection and optimization. The system generates experimental runs based on the chosen DoE model, considering factors such as the number of variables, their levels, and the requirements for replication. The design can focus on the most relevant formulation space by setting appropriate upper and lower limits, particularly for numerical factors, such as oil, surfactant, and co-surfactant concentrations. The pseudo-ternary phase diagram often serves as a reference for defining these concentration ranges, ensuring that only the stable nanoemulsion regions are explored. Software such as Design-Expert version 9/10/11/12/13, Systat version 13, Statgraphics® centurion XV version 15.2.05, and MODDE software version 2.1 can assist in generating experimental runs [128,129]. Referring to Table 4, Table 5 and Table 6, the number of runs used in the experimental designs varied widely, ranging from 8 to 32. However, the most commonly used run is between 13 and 17. The sertraline SNEDDS formulation was developed using a 2^3^ factorial design with eight experimental runs. Oil, surfactant, and co-surfactant concentrations were evaluated at two levels (low and high) without replication. This design choice ensures the efficient screening of the selected factors while maintaining a minimal number of runs [123]. However, this approach is limited in capturing nonlinear relationships and is not ideal for formulation optimization [77]. Venetoclax SNEDDS was optimized using 32 experimental runs with a Central Composite Design. Five factors were considered: the oil concentration, surfactant concentration, co-surfactant concentration, stirring rate, and stirring time. Four response parameters were measured: the globule size, PDI, emulsification time, and % transmittance. Many factors directly contributed to the higher number of experimental runs. Although this approach requires more effort, time, and resources, it allows a more comprehensive understanding of the formulation. This design provides valuable insights into CMAs and CPPs, leading to more precise optimization of the SNEDDS formulation [110]. Selecting an appropriate number of runs is crucial for balancing experimental efficiency and data accuracy, ensuring that only the most relevant factors are considered for analysis and optimization.

### 6.4. Preparation and Characterization of SNEDDS

This stage involved preparing SNEDDS formulations according to designated experimental points. API is typically incorporated alongside oil, surfactant, and co-surfactant, ensuring the required dosage to achieve the desired pharmacological effect. The formulation process involved mixing the components via vortexing, stirring, or ultrasonication. Additionally, the solubility of APIs in the oil phase or mixture must be ensured to achieve optimal dissolution. Once all formulations have been prepared, they are characterized using appropriate and validated methods based on predefined response parameters.

### 6.5. Data Analysis and Polynomial Modelling

DoE provides a systematic framework for analyzing the relationship between factors and responses within a given system. The process begins with data analysis, where experimental responses are statistically processed to identify trends, interactions, and significant factors. Statistical tools, such as ANOVA, assess factor significance and interactions, ensuring that the selected model accurately represents the system. Polynomial modelling establishes a mathematical relationship between factors and responses, choosing an appropriate polynomial model crucial for accurately describing the system [129].

The selection of a polynomial model depends on how well it represents experimental data. A first-order polynomial assumes a simple linear relationship between the factors and the responses, which is suitable when the initial data show no curvature. A higher-order model may be required if the residual analysis or statistical fit tests indicate poor model adequacy. A second-order polynomial (quadratic model) is applied when nonlinearity is detected, with significant quadratic terms (X^2^) indicating nonlinear factor–response interactions. It is preferable that this quadratic model significantly improves the lack-of-fit test and increases R^2^ without overfitting. A third-order polynomial (cubic model) may be required for more complex response patterns to capture higher-order interactions when the quadratic model is insufficient [111,130].

Ensuring the validity of the model requires statistical tests, including R^2^ and Adjusted R^2^ (goodness of fit), *p*-values from ANOVA (significance of model terms), lack-of-fit tests (model adequacy), and residual analysis (error distribution check). A well-chosen model should balance accuracy and simplicity, favouring the simplest model that sufficiently explains the data [128,129]. In one study, each response could follow a different model, depending on the characteristics of the data and the relationship between factors and responses. For example, four responses were analyzed to optimize the SNEDDS Voxelotor: globule size, PDI, emulsification time, and % transmittance. After model analysis, it was found that emulsification time and PDI followed a quadratic model, indicating a nonlinear relationship that second-order interactions could explain. The globule size and % transmittance follow the cubic model, reflecting more complex variations with higher-order interactions [94]. Another study using glimepiride as an API demonstrated different model behaviours. The analysis revealed that globule size followed a cubic model, whereas drug solubility followed a linear model, suggesting a direct proportional relationship between the factors and the response. This further reinforces the idea that each response may exhibit a distinct mathematical relationship, necessitating careful model selection to ensure an accurate optimization [89].

### 6.6. Analysis of the Relationship Between Factors and Responses

The relationships between the factors and responses were analyzed using the selected polynomial equation and contour plots. The polynomial equation provides insight into the influence of each factor on the response by analyzing the regression coefficients and assessing the model significance through ANOVA. Additionally, contour and response surface plots were used to visualise factor interactions and their effects on the response, providing a comprehensive understanding of the optimization process. Research data from various studies indicate that factor–response correlations can vary depending on experimental conditions and system characteristics. However, specific recurring patterns have frequently been observed regarding the impact of formulation factors on the SNEDDS responses (Table 4, Table 5 and Table 6). For example, the general influence of formulation factors on globule size, % transmittance, and emulsification time is explained.

Globule size in SNEDDS is primarily influenced by the concentration of oil, surfactant, and co-surfactant. Higher oil content tends to increase globule size due to the greater volume of the dispersed phase. In contrast, increasing the concentration of surfactants and co-surfactants typically reduces globule size by stabilizing the oil–water interface. An effective strategy to achieve smaller and more stable globules involves reducing the oil concentration while increasing the surfactant content. Surfactants lower interfacial tension and facilitate the spontaneous formation of finer emulsified droplets, thereby enhancing formulation stability and drug absorption [107,109].

The % transmittance is directly related to the clarity of the nanoemulsion system. Smaller globule sizes generally result in higher % transmittance, as they reduce light scattering and make the system appear more transparent system. Achieving transmittance closer to 100% typically requires reducing the oil concentration and/or increasing the surfactant and co-surfactant concentrations [102,106].

The PDI reflects the uniformity of droplet size distribution within a nanoemulsion system. A lower PDI value indicates a more homogeneous dispersion, which is desirable for formulation stability and reproducibility. PDI tends to increase with higher oil concentrations, as excessive oil may lead to the formation of larger and more variable droplet sizes. Conversely, decreasing the oil content while increasing the surfactant and co-surfactant levels can promote the formation of more uniform droplets, resulting in lower PDI values. This improvement in uniformity is primarily due to the enhanced stabilization of the dispersed phase provided by sufficient surfactant coverage at the oil–water interface [59].

The emulsification time is critical for the development of SNEDDS. Faster emulsification is desirable to ensure rapid nanoemulsion formation. The surfactant system primarily influenced the emulsification time. Higher surfactant levels accelerate emulsification by reducing interfacial tension. Higher oil concentrations, especially those with highly viscous oils, prolong the emulsification time. If the goal is to shorten the emulsification time, increasing the surfactant concentration, optimizing the surfactant-to-cosurfactant ratio, and selecting surfactants with appropriate HLB values can be effective strategies [108,119].

The relationship between formulation factors and response parameters such as globule size, PDI, and percent transmittance is not always generalizable. Different studies may yield varying trends depending on the specific composition and experimental conditions. Therefore, polynomial equation analysis is essential to quantitatively describe a given formulation space’s factor–response relationships. This statistical modelling allows for identifying significant variables, interaction effects, and optimal formulation regions, thereby supporting rational and efficient SNEDDS development.

### 6.7. Desirability Function Analysis and Validation

The desirability function is a standard tool for optimizing formulations to determine the best factor combination that simultaneously satisfies multiple responses. Each response obtained a desirability score (0–1), and a composite score reflected the overall optimization. The goal was to maximize the score by adjusting the formulation variables. After the optimal formulation was identified, a validation process was conducted to confirm the accuracy of model predictions. This involved preparing and testing the optimized formulation under the same experimental conditions and comparing the observed results with the predicted values. The model is reliable if the experimental data closely match the predicted responses. At the end of this stage, a fully validated optimal SNEDDS formulation was obtained, ensuring its robustness, reproducibility, and potential for further development [115,131].

## 7. In Vitro, Ex Vivo, and In Vivo Performances of Optimized SNEDDS

QbD and DoE were applied to SNEDDS development to obtain an optimized formulation with good in vitro and in vivo performance. As SNEDDS is generally developed for poorly water-soluble compounds, the primary goal of its development is to enhance solubility and % drug release, ultimately improving its pharmacokinetic profiles and therapeutic efficacy [42,132].

### 7.1. The Influence of SNEDDS on the Drug Release Profile

The primary objective of SNEDDS development is to enhance the dissolution or release of APIs in the gastrointestinal tract, which is typically evaluated using in vitro testing. As shown in Table 4, Table 5 and Table 6, the SNEDDS formulations successfully improved drug release for all APIs, although the degree of enhancement varied. These differences may be attributed to several factors, including the physicochemical properties and formulation composition of the drug. The enhancement dissolution in SNEDDS can be attributed to several key mechanisms. SNEDDS forms nanoemulsions with small globule sizes, increasing the surface area for drug–media interactions and accelerating drug release. The presence of surfactants and co-surfactants in the formulation reduces interfacial tension, enhances drug solubility, and prevents precipitation in gastrointestinal fluids. SNEDDS can maintain the drug in a solubilized state, preventing solubility limitations commonly observed with lipophilic compounds in the gastrointestinal tract [5,133].

Bosentan formulated in SNEDDS exhibited a significant increase in dissolution compared to conventional tablet formulations, with an enhancement of 3- to 7.97-fold, depending on the dissolution medium used [47]. A similar trend was observed with voxelotor, which showed a 3.1-fold increase in drug release compared to its pure form [94]. A significant enhancement was also observed in the atorvastatin/ezetimibe SNEDDS, with a drug release reaching approximately 99% compared to only 8% for the pure drug, highlighting the potential of SNEDDS to improve dissolution and oral bioavailability. Additionally, the in vitro release of zaleplon in SNEDDS was 17 times faster than that of the marketed tablet, further demonstrating the efficacy of SNEDDS in enhancing drug release rates. The active ingredient in these formulations is a BCS class II compound with low water solubility, which, through the development of SNEDDS, significantly improved dissolution and drug release [134,135]. A slightly different result was observed when developing the SNEDDS for rosuvastatin, indicating a lower degree of dissolution enhancement. The dissolution improvement of rosuvastatin was 1.11 times higher than that of the marketed tablets and 1.64 times higher than that of the pure drug. This phenomenon can occur because rosuvastatin is a hydrophilic statin, meaning that the solubility enhancement provided by SNEDDS is not as significant as in API with lower water solubility [91]. Other factors, such as lipid composition, surfactant-to-surfactant ratio, and compatibility with the dissolution medium, may also contribute to these differences.

### 7.2. The Influence of SNEDDS on Drug Permeation

In addition to enhancing drug release, the development of SNEDDS plays a pivotal role in improving drug permeation, a parameter typically evaluated through in vitro or ex vivo methodologies. In vitro permeability studies are commonly conducted using the Caco-2 cell monolayer model, which closely mimics the human intestinal epithelium, enabling a quantitative assessment of transcellular drug transport [136]. This model has been extensively applied in evaluating SNEDDS formulations. For instance, in vitro studies on palbociclib and letrozole revealed that SNEDDS formulations led to approximately 4-fold and 1.7-fold increases in apparent permeability (Papp) values, respectively [121]. Similarly, fosfestrol-loaded SNEDDS demonstrated a remarkable 4.68-fold enhancement in permeability across the Caco-2 monolayer [119]. Curcumin, another poorly soluble compound, exhibited a 3.44-fold increase in permeability when formulated as SNEDDS compared to its pure drug [103].

The permeability-enhancing capacity of SNEDDS has also been substantiated through ex vivo studies, which generally utilize isolated intestinal membranes from animal models, such as rats or rabbits. These models facilitate the evaluation of passive drug diffusion across biological membranes, closely resembling physiological conditions. For example, SNEDDS of plumbagin achieved a permeation of 90.36% ± 2.78%, significantly outperforming the conventional suspension formulation, which exhibited only 46.58% ± 2.10% permeation [107]. Additionally, SNEDDS containing bosentan resulted in a 3.36- to 16.6-fold increase in drug permeation compared to the reference tablet [47]. In another study, SNEDDS comprising neomycin sulfate and thioctic acid were assessed using rabbit intestinal membranes, demonstrating substantial enhancement in permeability over non-SNEDDS formulations [83]. Moreover, the use of a non-everted gut sac model further supports the permeability enhancement capability of SNEDDS. In the case of ibrutinib-SNEDDS, this model was employed to evaluate drug permeation across various segments of the gastrointestinal tract, including the duodenum, jejunum, ileum, and colon. Significant improvements in permeability were observed across all regions, underscoring the broad-spectrum efficacy of SNEDDS in facilitating the transport of drugs [90].

The observed improvements in both in vitro and ex vivo permeation can be attributed to multiple interrelated factors. Primarily, the increased solubilization of lipophilic drugs within the nanoemulsion matrix enhances their thermodynamic activity and availability at the absorption site. The nano-sized oil droplets in SNEDDS formulations provide a large interfacial surface area for drug diffusion, allowing for intimate contact with the intestinal epithelium and thereby promoting absorption. Furthermore, SNEDDS may facilitate drug transport via both passive diffusion and active vesicular mechanisms, such as pinocytosis or endocytosis, even under ex vivo experimental conditions. These attributes collectively contribute to the superior permeability profile of SNEDDS-formulated drugs [47].

### 7.3. The Influence of SNEDDS on Pharmacokinetic Profiles

The enhanced dissolution and permeation in SNEDDS formulations also influenced the pharmacokinetic profile. This is primarily due to increased solubility and drug release in the gastrointestinal tract, allowing for better absorption. In addition to improved dissolution, higher bioavailability can be attributed to several mechanisms, including enhanced permeation across the intestinal membrane, which facilitates greater drug transport into systemic circulation. Moreover, other contributing factors include lymphatic absorption, which bypasses first-pass metabolism in the liver, inhibition of P-glycoprotein (P-gp) efflux, and protection against enzymatic metabolism. These combined mechanisms contribute to higher plasma drug levels and an overall improvement in the pharmacokinetic profile [5,137,138].

Most studies on SNEDDS development include pharmacokinetic studies and have demonstrated an improved pharmacokinetic profile compared to pure drugs or conventional formulations. For example, SNEDDS formulated for resveratrol significantly enhanced pharmacokinetic parameters (48-fold for AUC_0–720_ and 16-fold for C_max_). The significant increases in AUC_0–720_ min and Cmax can be attributed to several factors. First, the smaller globule size of the emulsion enhanced surface interaction with enterocytes, improving intestinal permeability. Additionally, the hydrophilic outer layers of Kolliphor^®^ RH40 and TPGS facilitated diffusion across the unstirred water layer, a key barrier before absorption. Second, as resveratrol is a P-gp substrate, P-gp inhibitors such as TPGS and Kolliphor^®^ RH40 help reduce efflux and enhance absorption. Finally, lipid-based excipients, particularly TPGS, promote chylomicron secretion and lymphatic transport, thereby bypassing hepatic metabolism and significantly increasing their bioavailability [61,137,138].

A similar enhancement was observed in SNEDDS formulated for camptothecin, which showed a 17-fold increase in bioavailability. Like resveratrol, lymphatic absorption, the ability to bypass first-pass metabolism, and P-gp efflux are key factors in enhancing bioavailability. Rapid emulsification, increased surface area, and improved dissolution rate facilitate faster and more efficient drug absorption. Furthermore, lipids and surfactants may help retain SNEDDS at the apical membrane, thereby prolonging its residence time and enhancing its absorption and bioavailability [116]. The improvement in the pharmacokinetic profile due to the SNEDDS formulation has also been observed for other active compounds, as shown in Table 4, Table 5 and Table 6.

### 7.4. The Influence of SNEDDS on Therapeutic Efficacy

SNEDDS formulations have been developed for various active pharmaceutical ingredients with diverse therapeutic applications. Several studies have comprehensively evaluated SNEDDS in terms of drug release, pharmacokinetics, and therapeutic efficacy. Research on SNEDDS formulations of glimepiride has demonstrated that SNEDDS-based tablets exhibit superior efficacy compared to non-SNEDDS formulations and marketed tablets, as evidenced by a more significant reduction in blood glucose levels [89]. The development of SNEDDS containing alpha-lipoic acid (ALA) with gastroprotective activity also enhanced the pharmacodynamic profile. The optimized ALA-SNEDDS significantly reduced the gastric ulcer index compared to raw ALA, indicating improved therapeutic efficacy [86]. The development of SNEDDS for candesartan enhanced its efficiency compared to standard drug solutions and marketed tablets. The formulation significantly improved systolic blood pressure profiles in experimental animals, demonstrating superior in vivo performance based on pharmacodynamic parameters [40]. The development of SNEDDS for benidipine resulted in more significant therapeutic effects than those of the suspension form. The SNEDDS formulation significantly improved blood pressure parameters in hypertensive model animals [38]. SNEDDS for flufenamic acid exhibited significantly higher anti-inflammatory activity than the pure drugs after 6 h of oral administration. This improvement in therapeutic efficacy was attributed to the presence of lipids and surfactants, which enhanced solubility and membrane permeability, along with various other factors contributing to the increased bioavailability of flufenamic acid in SNEDDS formulations [118].

Various studies have demonstrated that SNEDDS formulations can enhance therapeutic efficacy compared with pure drugs or marketed formulations. SNEDDS are primarily used for oral drug delivery, particularly for API with systemic effects. An increase in bioavailability is positively correlated with therapeutic efficacy. Further research is required to advance the development of SNEDDS in clinical trials.

## 8. In Silico Approach in SNEDDS Development

Applying an in silico approach in the development of SNEDDS has revolutionised formulation optimization by enabling a more rational and efficient approach to system design. However, its application remains relatively limited in current pharmaceutical research. The ability to model molecular interactions and phase behaviour in silico allows a deeper understanding of how different excipients interact in a nanoemulsion system. Computational methods have become essential in drug formulation, particularly for poorly water-soluble drugs that require enhanced solubilization strategies. SNEDDS has emerged as a practical approach to improve drug absorption by forming nano-sized emulsions upon administration, and computational techniques help optimize their composition. Before conducting experimental trials, researchers can predict formulation properties through machine learning algorithms and molecular simulations. Furthermore, computational models offer insights into the structural stability of nanoemulsions, enabling the rational selection of excipients. With the growing complexity of drug formulations, in silico modelling provides a pathway to streamline the development process and improve the efficiency of SNEDDS formulations [139,140].

The development of SNEDDS formulations based on in silico approaches is still relatively limited (Table 7). However, alongside advancements in computational technology and formulation prediction tools, the trend of utilizing in silico approaches is expected to continue increasing. One study reported the development of meloxicam SNEDDS by integrating machine learning and in silico modelling. Random Forest (RF)-based machine learning was employed to design the pseudo-ternary phase diagram, yielding a system with excellent predictive performance, achieving a prediction accuracy of 89.51%. RF works by building multiple decision trees, each trained on a random subset of data and features. The final prediction is made by majority voting across all trees, making the model more accurate and robust [141]. Prediction of the pseudo-ternary phase diagram successfully identified the optimal combination of components, consisting of oil (Labrafil M 1944 CS), surfactant (Cremophor RH40), and cosurfactant (Transcutol HP). The optimum formulation was determined experimentally using a Central Composite Design. This formulation was then further evaluated through molecular dynamic simulations. The simulation box, containing drug molecules, excipients, and water as the solvent, was constructed using the PACKMOL package ( The MD simulations were conducted using the AMBER 18 software package, with molecular interactions described by the General Amber Force Field (GAFF). After 200 ns of simulation, the oil and surfactant molecules formed the core structure, or droplet skeleton, of the Meloxicam-SEDDS, while the cosurfactant molecules were distributed around the droplet surface. The radius of gyration (Rg) and solvent-accessible surface area (SASA) of the system containing the cosurfactant were both higher than those of the cosurfactant-free system. These findings indicate that the presence of cosurfactant enhances the self-emulsification capacity of the system in aqueous media. The simulation results were in good agreement with experimental observations, further demonstrating that the cosurfactant plays a crucial role in enhancing the emulsification performance of SEDDS [19]. These results highlight the synergistic role of molecular simulations and machine learning in optimizing SNEDDS formulations, enabling researchers to design stable and effective nanoemulsions [142].

A few studies have explored the in silico optimization of self-assembly nanoemulsions, which could potentially be applied to developing SNEDDS formulations. Self-assembly nanoemulsions are closely related to SNEDDS because both systems rely on the spontaneous formation of nanometer-sized emulsions when exposed to an aqueous environment. In this context, self-assembly is a spontaneous process in which constituent molecules autonomously arrange and organize themselves into a defined structure without requiring external energy inputs. In silico-based studies to determine the optimum formulation of self-assembly nanoemulsions have been conducted for various systems, including palm kernel oil wax esters, black cumin oil, soybean oil-curcumin, and vitamins A/E [143,144,145,146]. Various studies have analyzed the interactions between oil, surfactant, and cosurfactant in nanoemulsion systems, which is commonly conducted using the GROMACS. These studies utilize a variety of force fields, such as OPLS-AA, GROMOS96, and MARTINI, depending on the required level of molecular detail and the simulation’s complexity [143,144,145,146].

An in silico approach has been conducted to observe the self-assembly process of nanoemulsion formed between palm kernel oil wax esters (PKOEs) and the surfactant (Tween 80). This simulation aims to evaluate the system’s stability and to understand the molecular mechanisms underlying nanoemulsion formation. The PKOEs-Tween 80 system formed a prolate ellipsoid micelle, where the Rg decreased from 3.92 nm to 3.08 nm. The transition to a more compact structure over time suggests that surfactant-mediated interactions are crucial in micelle stabilization. These findings indicate that self-assembly dynamics play a key role in determining the stability and morphology of SNEDDS formulations [144].

A similar approach has also been applied to investigate the self-assembly process of nanoemulsions containing black cumin oil (BCO). A BCO-surfactant study showed that BCO-Tween 20 and BCO-Lecithin formed stable spherical micelles, while BCO-Span 20 and BCO-Span 80 failed to self-assemble. These findings suggest that selecting the appropriate surfactant is crucial in achieving a stable nanoemulsion system, as only specific surfactant–oil combinations result in successful micelle formation. The study also demonstrates that the MD simulation has the potential to identify suitable surfactants capable of spontaneously forming a stable nanoemulsion [145].

MD simulation was also used to analyze the molecular behaviour of nanoemulsions containing soybean oil, with and without curcumin. The results showed that curcumin affected nanoemulsion droplets’ size, shape, and internal arrangement. Particle sizes ranged from 2 to 6.3 nm, with assemblies exhibiting spherical and prolate spheroid shapes consistent with experimental data. Curcumin is consistently localized at the surface of the assemblies, while other molecules are arranged with hydrophobic parts inside and hydrophilic parts at the surface [146].

MD simulation has also been applied to investigate the self-assembly process of nanoemulsions containing Benzalkonium chloride (BZK) as the surfactant, cyclohexane as the oil phase, and ethanol as the co-surfactant in an aqueous medium. The BZK nanoemulsion exhibited a prolate ellipsoid structure with an Rg of 1.68 nm, emphasizing the role of surfactants in droplet stabilization [147]. These findings highlight the importance of electrostatic and hydrophobic interactions in determining the structural integrity of nanoemulsions. Predicting micelle formation behaviour using molecular descriptors allows for better formulation control. Computational techniques offer insights into droplet morphology, which directly influences drug release rates and bioavailability.

Based on the previous discussion, in silico modelling serves as a critical tool in SNEDDS development, elucidating the mechanisms behind self-assembly, molecular interactions, and emulsification processes. By combining molecular simulations with machine learning techniques, a more targeted and efficient formulation design is facilitated, allowing for the identification of optimal and stable component mixtures that enhance performance.

**Table 7 pharmaceutics-17-00701-t007:** In silico approach in SNEDDS development.

Active Ingredient [Ref]	Components	Computational Method Used	Results Obtained
Meloxicam [19]	Labrafil M 1944 CS (oil), Cremophor RH40 (surfactant), and Transcutol HP (co-surfactant)	- The RF model was trained on 4495 SEDDS formulation datasets for predicting pseudo-ternary phase diagrams. - MD simulations were conducted using the AMBER 18 software with the GAFF force field. - The system was built using PACKMOL, and simulations ran for 200 ns at 310K.	- Machine learning successfully predicted the pseudo-ternary phase diagram with 89.51% accuracy. - RMSD analysis showed a more stable system with cosurfactants. - The Rg and SASA analysis confirmed efficient emulsification and dispersion in water.
Palm Kernel Oil Wax Esters (PKOEs) [144]	PKOEs (oil) and Tween 80 (surfactant)	- MD simulations were conducted using GROMACS 3.3.2 with the OPLS-AA force field. - The system run was performed in a 1000 nm^3^ cubic simulation box using the SPC water model. - Energy minimization was performed using the steepest descent method, followed by 1 ns NPT equilibration (300K, 1 bar) and a 20 ns production run in an NVT ensemble with a 2 fs time step. - The Particle Mesh Ewald (PME) method was employed for electrostatics, and van der Waals interactions were truncated at a cutoff distance of 0.9 nm.	- PKOEs and Tween80 molecules self-assembled into a stable prolate ellipsoid micelle. - Rg decreased from 3.92 nm to 3.08 nm, with the most compact structure observed between 16 and 17 ns (Rg = 2.49 nm). - The aggregate diameter was 5.78 ± 0.05 nm, consistent with experimental values (6.03 ± 0.05 nm). - SASA analysis revealed an initial total of 491.40 ± 3.80 nm^2^, which decreased significantly within the first 3 ns, stabilizing at 202.40 ± 1.20 nm^2^ around 18 ns.
Black Cumin Oil (BCO [145]	Tween 20, Tween 80, Span 20, Span 80, and Lecithin in equal proportions with 10 BCO molecules	- MD simulations were performed using GROMACS 2021 with the GROMOS96 force field. Initial configurations were generated using PACKMOL. - Energy minimization was done using the steepest descent algorithm (500,000 steps). The system was equilibrated under NVT (310K, 500 ps) and NPT (1 bar, 500 ps) ensembles. - The 50 ns production run was performed using a two fs time step, with electrostatics handled by the PME method (1.4 nm cutoff) and van der Waals interactions truncated at 1.4 nm. - Structural analysis was performed using RMSD, Rg, and SASA.	- BCO-Tween 20 and BCO-Lecithin formed stable spherical micelles with effective radii of 10.20 nm and 8.67 nm, respectively. BCO-Tween 80 formed two micelle clusters, while BCO-Span 20 and BCO-Span 80 failed to form micelles. - RMSD and Rg analyses indicated that BCO-Tween 20 maintained stability. - Enthalpy calculations confirmed that micellization was an exothermic, spontaneous process driven by electrostatic and hydrophobic interactions.
Benzalkonium chloride [147]	Cyclohexane (oil), Benzalkonium chloride (surfactant), and Ethanol (cosurfactant)	- GC-MD simulations were performed using GROMACS 4.5 with the MARTINI force field. - The system contained 51 BZK surfactants, 86 cyclohexane molecules, and 630 ethanol molecules in a periodic cubic simulation box. - Energy minimization was conducted for 100 ps, followed by NVT equilibration (310K, 50,000 steps) and NPT equilibration (1 bar, 500 ps). - The production simulation was performed for 1 μs using a 10 ps time step with a van der Waals interaction cutoff of 1.2 nm.	- Oil molecules accumulated in the core, while BZK surfactant chains stabilized the droplet surface. Ethanol molecules did not integrate into the droplet but formed a thin stabilizing layer. - The Rg of the nanoemulsion was 1.68 nm, and the estimated physical radius was 2.17 nm. - Shape analysis confirmed a prolate ellipsoid structure with an average eccentricity of 0.57. - The potential energy analysis showed rapid convergence, indicating a thermodynamically stable nanoemulsion.
Curcumin [146]	Soybean oil-tween 80 in two conditions: without curcumin (OL: TW80:H_2_O = 1.67:15:83.33) and with curcumin (OL:TW80:H_2_O: CUR = 0.17:1.66:15:83.19:0.17)	- MD simulations were conducted using GROMACS with the GROMOS96 force field. - The system was constructed based on experimental ratios and simulated in a periodic cubic box. - Energy minimization was performed, followed by NVT and NPT equilibration. The production run lasted 50 ns at 300 K. - Structural properties were analyzed using Rg, principal moments of inertia, and molecular distribution within aggregates.	- MD simulation shows the spherical and prolate spheroid-shaped aggregates (2 to 6.3 nm), with curcumin molecules preferentially localized at the surface of aggregates. - The presence of curcumin accelerated equilibrium formation and produced more compact and symmetric aggregates. - The Rg analysis confirmed structural stability and the organization of hydrophobic and hydrophilic regions.

## 9. Challenges and Limitations

While QbD and DoE have been increasingly adopted in SNEDDS formulation, several practical limitations remain, particularly when considering their broader implementation in real-world and industrial settings. A common challenge is the variability in the design. Some reported studies utilize relatively simple optimization designs, which may not fully account for complex interactions between formulation components. Furthermore, inconsistencies in experimental conditions, such as emulsification techniques, dilution ratios, or digestion models, can hinder reproducibility and make it challenging to draw generalized conclusions across studies.

Another notable limitation is the insufficient validation of optimized formulations under conditions relevant to scale-up and long-term stability. Although the QbD framework emphasizes the definition of a design space and the identification of CPPs, many studies predominantly focus on formulation-related variables, such as the types and ratios of oils, surfactants, and co-surfactants. This limited incorporation of CPPs reduces the ability to predict formulation behaviour during scale-up, where factors such as mixing speed, emulsification method, and temperature control may have a significant impact on product quality and consistency. In the context of industrial manufacturing, while the QTPP and CQAs generally remain consistent with those established during early development, the most critical adjustments are typically required in CPPs. These modifications are essential to ensure process reproducibility and product robustness at larger production scales. Furthermore, CMAs may also require refinement to accommodate variability in raw materials across production batches. The lack of such considerations in early stage studies highlights a significant gap in translating QbD-optimized SNEDDS formulations from the laboratory to industrial settings.

Regarding in silico modelling, while this approach provides valuable mechanistic insights and predictive capabilities, its application in SNEDDS development is still evolving. Limitations include the availability of high-quality input data, the need for system-specific model validation, and the expertise required to interpret complex simulation results. Additionally, integrating in silico approaches with QbD frameworks remains an area of ongoing exploration.

## 10. Conclusions

The successful development of SNEDDS is a complex process due to the intricate interrelationships among its key components (oil, surfactant, and co-surfactant), which significantly influence formulation stability, emulsification efficiency, and drug-loading capacity. QbD provides a systematic, science-driven framework for addressing these challenges effectively. By integrating DoE, QbD facilitates the quantitative evaluation of CMAs and CPPs and their impact on CQAs. This approach enables the optimization of component ratios in SNEDDS, reduces variability, and enhances overall product robustness. The application of DoE, such as mixture design, response surface methodology, and factorial design, not only deepens process understanding but also supports the development of SNEDDS with improved drug release, enhanced permeability, and favorable pharmacokinetic profiles. In silico modelling plays a pivotal role in the development of SNEDDS by offering deep insights into self-assembly dynamics, molecular interactions, and emulsification behaviour. By integrating molecular simulations with machine learning, this approach enables rational and efficient optimization of formulations through the identification of stable and effective component combinations.

## 11. Future Perspectives

The progression of SNEDDS development through a QbD framework offers a rational and efficient pathway for formulating poorly water-soluble drugs. Nevertheless, to translate optimized formulations into clinical and commercial success, further research is essential in the post-formulation stages. Key areas include formulation validation under physiologically relevant conditions, scale-up manufacturing, long-term stability assessment, and comprehensive preclinical and clinical evaluations to confirm safety and efficacy. Additionally, compliance with regulatory standards and thorough documentation—including risk assessment, design space justification, and process control strategies—remains fundamental for achieving marketing authorization, especially for complex delivery systems like SNEDDS.

In this context, special attention should also be directed toward addressing the unique needs of vulnerable populations, particularly pediatric and geriatric patients, who often face challenges such as swallowing difficulties, variable gastrointestinal physiology, polypharmacy, and altered metabolic capacities. The versatility of SNEDDS—characterized by ease of administration, potential for taste masking, and the capability to improve bioavailability without relying on bile salt-dependent solubilization—makes it a promising platform for these populations. Pediatric applications may benefit from SNEDDS in the form of mini-capsules, oral suspensions, or reconstitutable powders. In geriatric care, SNEDDS offers an opportunity to reduce pill burden, enhance the absorption of drugs compromised by age-related gastrointestinal changes, and support better therapeutic adherence. However, to ensure optimal translation, age-specific formulation design should incorporate biorelevant dissolution studies, palatability assessments, and patient-centric delivery formats [148].

Beyond the QbD paradigm, emerging digital technologies—such as artificial intelligence (AI), machine learning (ML), in silico modelling, and physiologically based pharmacokinetic (PBPK) simulation—are becoming increasingly integral to SNEDDS development. These tools not only complement traditional experimental approaches but also provide mechanistic insights and predictive capabilities that can significantly streamline formulation processes. While machine learning techniques, such as Random Forest, have demonstrated success in predicting optimal excipient combinations, as exemplified in meloxicam-loaded SNEDDS, future advancements are expected to shift towards more integrative computational approaches [19]. In particular, PBPK modelling has gained traction as a powerful tool for simulating the absorption, distribution, metabolism, and excretion (ADME) of drugs formulated in SNEDDS [149]. By incorporating physiological, biochemical, and physicochemical parameters into mathematical frameworks, PBPK models enable the prediction of in vivo pharmacokinetics [150]. For instance, PBPK simulation of sildenafil-loaded SEDDS demonstrated a 28% increase in bioavailability, reinforcing the critical impact of droplet size and formulation attributes on systemic drug exposure [151]. In parallel, the adoption of AI-driven platforms such as FormulationAI offers a transformative shift in SNEDDS optimization. These systems utilize large datasets and predictive algorithms to estimate critical formulation parameters, including droplet size, polydispersity index (PDI), and encapsulation efficiency, thereby reducing experimental workload and expediting development timelines. By integrating molecular descriptors and insights from molecular dynamics and PBPK simulations, AI-based platforms facilitate a more rational, cost-effective, and personalized approach to formulation design [152,153,154].

## Figures and Tables

**Figure 1 pharmaceutics-17-00701-f001:**
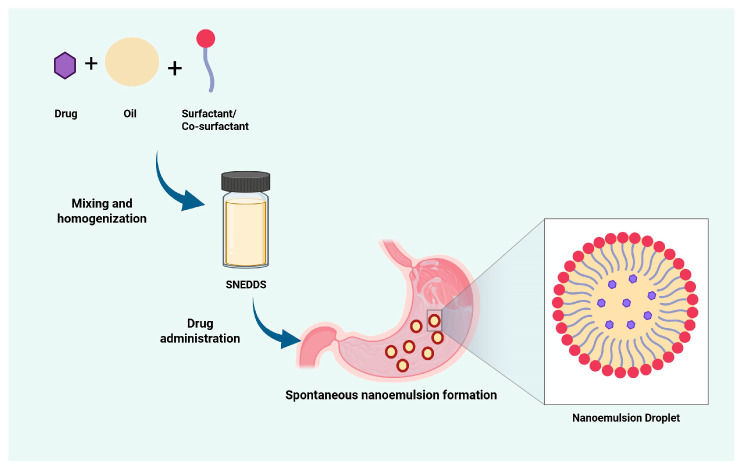
Illustration of SNEDDS (created in BioRender https://BioRender.com/e1zgkjm accessed on 23 May 2025).

**Figure 2 pharmaceutics-17-00701-f002:**
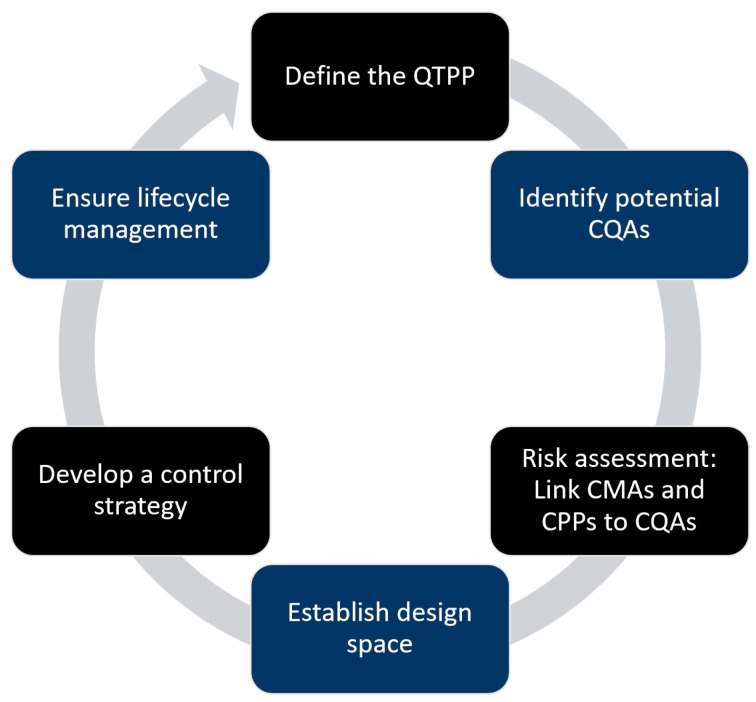
QbD stages based on ICH Q8 (R2).

**Figure 3 pharmaceutics-17-00701-f003:**
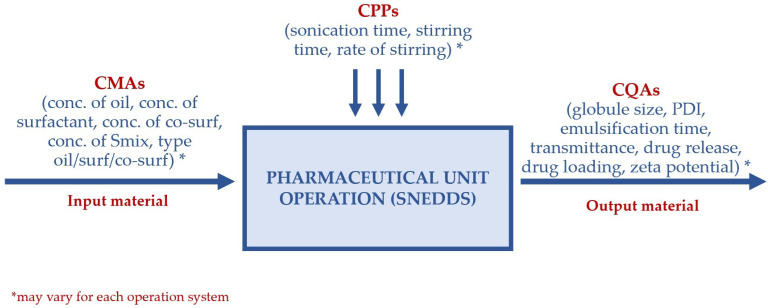
Relationship between CMAs, CPPs, and CQAs in SNEDDS optimization.

**Table 1 pharmaceutics-17-00701-t001:** Comprehensive QbD implementation in SNEDDS development.

API [Ref]	QTPP Elements	Risk Assesment Method	CQAs	CPPs	CMAs
Ritonavir [56]	Dosage type, dosage strength, route of administration, packaging, pharmacokinetic parameter, stability	Ishikawa fish-bone diagram follows with REM	Globule size, emulsification time, PDI, and % transmittance	-	Conc. of oil, surf, and co-S
Bedaquiline [57]	Dosage form, dosage type, dosage strength, route of administration, stability, container closure system, alternative method for administration	Ishikawa fish-bone diagram	Globule size, PDI, % transmittance	Sonication time	Conc. of oil and Smix
Benidipine [38]	Dosage form, dosage type, dosage strength, route of administration, pharmacokinetics, packaging, container closure system, different methods of administration, stability	Ishikawa fish-bone diagram,	Emulsification time, globule size, % drug release, % transmittance	-	Conc. of oil, surf, and co-S
Cinacalcet HCl [58]	Dosage form, dosage type, drug absorption, dispersity	Ishikawa fish-bone diagram followed by FMEA	% drug release, emulsification time, globule size, PDI	-	Conc. of oil, surf, and co-S
Docetaxel [59]	Drug delivery system, dosage type, route of administration, and drug release	Ishikawa fish-bone diagram	Globule size, PDI, % transmittance, emulsification time	Sonication time	Conc. of oil and Smix
Olmesartan medoxomil [60]	Dosage form, dosage type, dosage strength, route of administration, pharmacokinetics, packaging, stability	Ishikawa fish-bone diagram followed by RAM	Globule size, emulsification time, % drug release, mean dissolution time, % dissolution efficiency	-	Conc. of oil, surf, and co-S
Resveratrol [61]	Clinical target, route of Administration, dosage form design, stability, container closure system	RAM	Emulsification time, globule size, PDI, % drug release	-	Conc. of oil, surf, and co-S
Tamoxifen and Resveratrol [62]	Dosage form, dosage type, route of administration, stability	Ishikawa fish-bone diagram	Globule size, PDI, % transmittance	-	Conc. of oil and Smix

**Table 2 pharmaceutics-17-00701-t002:** QTPP in SNEDDS development [38,56,57,58,59,60,61,62].

QTPP Element ^1^	Target	Justification
Clinical Target	Improved bioavailability of poorly water-soluble drugs	SNEDDS enhances drug solubility and absorption by forming nanoemulsions in the gastrointestinal tract.
Dosage Form	SNEDDS/Lipid-based drug delivery system	SNEDDS is a lipid-based drug delivery system that can enhance the bioavailability of poorly water-soluble compounds with good stability.
Dosage Type	Immediate release	A quicker onset of action results in improved therapeutic effects.
Dosage Strength	Defined on drug solubility and therapeutic dose	The strength must ensure an optimal dose that achieves the desired pharmacokinetic profile.
Route of Administration	Oral	SNEDDS is designed for oral delivery to enhance drug absorption in the gastrointestinal tract.
Packaging/Container Closure System	Soft or hard capsules/airtight glass bottles	Protects the formulation from environmental factors and prevents drug–lipid interactions
Pharmacokinetic Parameters	Increased Cmax and AUC compared to conventional formulations	SNEDDS improves drug dissolution, leading to enhanced systemic exposure and faster onset of action.
Stability	Compliance with ICH guidelines	Ensures that the formulation remains effective and does not degrade under storage conditions

^1^ Depends on study condition.

**Table 3 pharmaceutics-17-00701-t003:** CQAs in SNEDDS development [38,56,57,58,59,60,61,62].

CQAs ^1^	Target	Justification	Method
Globule Size	<200 nm	Enhances drug absorption and bioavailability	Dynamic Light Scattering of the dilution system
Emulsification Time	<1 min	Ensures rapid self-nanoemulsification in the GI tract	Visual observation with gentle agitation in aqueous media
% Transmittance	>90%	Indicates a clear and stable nanoemulsion	UV-Vis spectrophotometry of the dilution system
Drug Release	~100 %(the limit varies for each API)	Ensures efficient and rapid drug release	Dissolution testing in appropriate media
Zeta Potential	±30 mV or higher	It prevents globule aggregation and enhances stability.	Electrophoretic Light Scattering of the dilution system
Polydispersity Index	<0.5	Indicates uniform globule size distribution	Dynamic Light Scattering of the dilution system

^1^ Depends on study condition.

**Table 4 pharmaceutics-17-00701-t004:** SNEDDS optimization using mixture design.

API [Ref]	Experimental Design	Correlation Factors and Responses	Optimal Formulation	Product Performances
Glimepiride + Rosuvastatin [80]	DoE: extreme vertices MD Runs: 13 Factors: % oil, surf, Co-S Responses: globule size	GS ↑: Oil and surf ↑, Co-S ↓	Oil: Curcuma longa oil (15%) Surf: Tween 80 (10%) Co-S: PEG 400 (75%) Characteristics: GS of 94.43 ± 3.55 nm and PDI of 0.544	- SNEDDS increased the % drug release compared to the marketed tablet. - SNEDDS increased Cmax and AUC with relative bioavailability 159.50% (GLM) and 245.16% (RSV) compared to the marketed tablet.
Febuxostat [81]	DoE: extreme vertices MD Runs: 14 Factors: % oil, surf, Co-S Responses: globule size, Stability Index	GS ↑: Oil and Co-S ↑, Surf ↓ Stability index ↑: Oil and Surf ↓, Co-S ↑	Oil: Corn oil (10%) Surf: Labrasol (40%) Co-S: Transcutol HP (50%) Characteristics: GS of 175.7 nm	- SNEDDS exhibited a significantly higher drug release compared to the marketed tablet (75% vs. 40%) - SNEDDS enhanced pharmacokinetic parameters, including an increase in Cmax and AUC with relative bioavailability of 146.4% compared to the marketed tablet.
Diclofenac potassium and coconut oil [82]	DoE: simplex lattice design Runs: 13 Factors: % of oil, surf, co-S Responses: globule size, PDI	GS ↑: Oil and Co-S ↑, Surf ↓ PDI ↑: Oil ↑	Oil: Coconut oil (10%) Surf: Tween 80 (70%) Co-S: Ethanol (20%) Characteristics: GS of 160 ± 7.5 nm, PDI of 0.380 ± 0.06, ZP: −38.2 ± 1.9 mV	- SNEDDS demonstrated a significant increase in drug release, achieving 80% release. - SNEDDS enhanced the pharmacological effect, with coconut oil improving gastric protection against ulcers induced by diclofenac potassium.
Neomycin Sulfate–Thioctic acid [83]	DoE: simplex lattice design Runs: 11 Factors: % oil, surf, co-S Responses: globule size	GS ↑: Oil ↑, Surf and Co-S ↓	Oil: Eucalyptus oil (13%) Surf: Tween 80 (46%) Co-S: Propyleneglycol (39%) Characteristics: GS of 150 nm	- SNEDDS demonstrated enhanced ex vivo intestinal permeation, as evidenced by sustained drug release over 12 h. - SNEDDS significantly enhanced the pharmacological effect, contributing to the suppression of hepatotoxic side effects.
Valsartan [84]	DoE: simplex lattice design Runs: 16 Factors: % oil, surf, co-S Responses: globule size; % drug load	GS ↑: Oil ↑, Surf and Co-S ↓ DL↑: Oil ↑	Oil: Sesame oil (24.9%) Surf: Tween 80 (33.3%) Co-S: PEG 400 (41.8%) Characteristics: GS of 174.6 nm, PDI of 0.184, and a ZP of 31.2 mV	- SNEDDS achieved a 97.33% drug dissolution within 10 min. - SNEDDS markedly improved pharmacokinetic performance, with increased Cmax and AUC and a relative bioavailability of 173.2% compared to the marketed tablet.
Tamoxifen [85]	DoE: mixture design Runs: 16 runs Factors: % oil, surf, co-S Responses: globule size, PDI, zeta potential, % drug release	NA	Oil: Corn oil (34%) Surf: Labrasol (48%) Co-S: Transcutol (18%) Characteristics: GS of 138 nm. PDI of 0.31, and ZP of +35.45 mV	- SNEDDS demonstrated a 77.21% drug release within 30 min. - SNEDDS enhanced cytotoxicity. - SNEDDS significantly improved pharmacokinetics, with a 375% increase in Cmax and a 391% increase in AUC compared to pure drugs.
Alpha-lipoic acid [86]	DoE: D-optimal mixture Runs: 16 Factors: % oil, surf, co-S Responses: globule size	GS ↑: Oil ↑	Oil: Pumpkin oil (10%) Surf: Tween 80 Co-S: PEG 200 Characteristics: GS of 97.12 nm	- SNEDDS significantly enhanced the pharmacological effect, with a marked improvement in the gastric ulcer index compared to the pure drug.
Artemisin [87]	DoE: optimal mixture design Runs: 16 Factors: % oil, surf, co-S Responses: % drug loading, solution, emulsification time	DL↑: Co-S ↓ Solution ↑: Oil ↑ ET ↑: Oil ↓, Surf and Co-S ↑	Oil: Labrafil M 1944 (50%) Surf: Cremophor EL (20%) Co-S: Transcutol P (30%) Characteristics: ET of 231 s, GS of 128.0 nm, ZP of −4.29 mV	- SNEDDS improved the solubility of the drug in vitro by 33.85-fold compared to the pure drug. - SNEDDS enhanced Cmax and AUC, resulting in a relative bioavailability of 147% compared to the pure drug.
Atorvastatin and ezetimibe [39]	DoE: D-optimal mixture design Runs: 16 Factors: % oil, surf, co-S Responses: globule size, zeta potential, % drug release, PDI	NA	Oil: Capryol 90 (10%) Surfactant: Tween 80/Koliphor RH 40 (42.71%) Co-S: Transcutol HP (47.29%) Characteristics: GS of 101.3 ± 0.47 nm; ZP of 23 mv, PDI of 0.241, dispersibility: grade A	- SNEDDS markedly enhanced drug release, achieving a 99% release compared to only 8% for the pure drug. - SNEDDS increased Cmax and AUC by 3.55- and 3.77-fold compared to the pure drug. - SNEDDS also improved the pharmacological effect by enhancing total cholesterol (TC) and non-HDL cholesterol levels compared to the suspension.
Candesartan [40]	DoE: D-optimal mixture design Runs: 14 Factors: % oil, surf, co-S Responses: globule size, % drug release, self-emulsification time	GS ↑: Oil ↑, surf and Co-S ↓ %DR ↑: Oil ↓, surf and Co-S ↑ ET ↑: Oil ↑, surf and Co-S ↓	Oil: Capmul PG-8 (5%) Surfactant: Kolliphor EL (32%) Co-S: Transcutol P (63%) Characteristics: GS of 13.91 nm, ET of 16s, ZP of 0.32 mV	- SNEDDS increased drug release by 1.99-fold compared to the pure drug and 1.10-fold compared to the marketed tablet. - SNEDDS enhanced the pharmacological effect by lowering systolic blood pressure more effectively than the marketed and pure drug.
Cefixime [41]	DoE: D-optimal design Runs: 20 Factors: % oil, surf, co-S Responses: globule size	GS ↑: Oil ↑, surf ↓	Oil: Cinnamon oil (40%) Surfactant: Tween 80 (40%) Co-S: PEG 200 (20%) Characteristics: GS of 130.73 ± 19.39 nm, PDI of 0.26 ± 0.01 nm, ZP of −9.50 ± 1.76 mV	- SNEDDS achieved a 77.26% drug release. - SNEDDS increased Cmax and AUC, with a relative bioavailability of 205% compared to the pure drug.
Exenatide [88]	DoE: D-optimal design Runs: 15 Factors: % MCT, MGDG, surf Responses: globule size	GS ↑: MCT ↑, MGDG and surf ↓	Oil: Captex 300/Capmul MCM (70%) Surfactant: Kolliphor RH 40 (30%) Characteristics: 26 ± 4 nm (Kolliphor rich), 231 ± 8 (MCT Rich)	- SNEDDS demonstrated increasing drug permeation base on in vitro permeability studies in Caco-2 cells. - SNEDDS enhanced Cmax and AUC, with formulations containing higher levels of MGDG and Kolliphor^®^ RH40 showing greater drug permeation and bioavailability than those with higher MCT
Glimepiride [89]	DoE: mixture design Runs: 17 runs Factors: % oil, surf, co-S Responses: globule size, solubility	GS ↑: Oil ↑, surf ↓ solubility ↑: surf↑	Oil: Black seed oil (15%) Surfactant: Tween 80 (40%) Co-S: PEG 400 (45%) Characteristics: GS of 34.64 nm and solubility of 36.67%	- SNEDDS showed a drug release range of 75.15% to 109.47%. - SNEDDS enhanced Cmax and AUC, with a relative bioavailability of 172% compared to the non-SNEDDS formulation and 106% compared to the marketed tablet. - SNEDDS improved pharmacological effects by lowering blood glucose levels more effectively than the non-SNEDDS formulation and the marketed tablet.
Ibrutinib [90]	DoE: I-optimal mixture Runs: 16 Factors: % oil, surf, co-S Responses: globule size, PDI, drug loading	GS ↑: Oil ↑, surf and Co-S ↓ PDI ↑: Oil ↑, surf and Co-S ↓ DL ↑: Oil ↑, surf and Co-S	Oil: Eugenol (5%) Surfactant: Tween 80 (80%) Co-S: PEG 200 (15%) Characteristics: GS of 60.85 nm, PDI of 0.195, ZP −11.9 mV, ET: 7 s	- SNEDDS showed significantly faster and more complete drug release in gastric and intestinal fluids compared to the pure drug. - SNEDDS enhanced drug permeation significantly. - SNEDDS increased Cmax and AUC, with a 3.86-fold increase in bioavailability in the fasted state compared to the free drug.
Resveratrol [61]	DoE: mixer design Runs: 13 Factors: % oil, surf, co-S Responses: globule size, PDI, emulsification time, % drug release	GS ↑: Oil ↑, surf and Co-S ↓ PDI ↑: Surf ↑, Co-S ↓ ET↑: Oil, surf, co-S ↑ %DR: co-S ↑	Oil: Labrafil^®^ M 1944 (23%) Surfactant: Kolliphor RH 40 (41%) Co-S: Transcutol HP (36%) Characteristics: GS of 22.07 nm, PDI of 0.148 nm, ET of 21.67 s	- SNEDDS showed a 83.93% drug release. - SNEDDS increased Cmax and AUC, with a 48-fold increase in bioavailability compared to the pure drug.
Rosuvastatin [91]	DoE: D-optimal mixture design Runs: 14 Factors: % oil, surf, and co-S Responses: globule size, % drug release, emulsification time	GS ↑: Oil and Co-S ↑, Surf ↓ ET↑: Oil ↑, surf ↓ %DR↑: Oil ↓, surf ↑	Oil: Capmul MCM EP (14%) Surfactant: Tween 20 (50%) Co-S: Transcutol P (36%) Characteristics: GS of 14.69 nm, PDI < 0.5, ZP of −4.09 mV	- SNEDDS achieved a 98.52 ± 0.05% drug release, higher than the marketed tablet and the pure drug. - SNEDDS demonstrated better pharmacological efficacy in lowering LDL and total cholesterol, while enhancing HDL levels, compared to both the marketed and pure drugs.
Sildenafil Citrate [92]	DoE: mixture design Runs: 16 Factors: % of oil, surf, and co-S Responses: globule size	GS ↑: Cons of Oil ↑, Surf ↓	Oil: Clove oil/oleic acid (10%) Surfactant: Tween 20 (60%) Co-S, Propylene glycol (30%) Characteristics: GS of 103.5 nm	- SNEDDS showed increased drug release. - SNEDDS enhanced Cmax and AUC, with a 1.44-fold increase in bioavailability compared to the marketed tablet.
Voxelotor [94]	DoE: D-optimal mixture design Runs: 16 Factors: % of oil, surf, and co-S Responses: globule size, PDI, emulsification time	GS ↑: Oil ↑, Surf and Co-S ↓ PDI ↑: Oil ↑, Surf ↓	Oil: Capryol PGMC (40%) Surfactant: Cremophor-E L (43%) Co-S: Labrafil M 1944 (17%) Characteristics: GS of 34.9 nm, PDI of 0.2, ZP of −8.4 mV, and ET of 32.4 s	- SNEDDS showed a 87% drug dissolution, which was significantly better than the pure drug (29%). - SNEDDS enhanced drug permeation with a 22-fold higher apparent permeability coefficient compared to the pure drug. - In vivo, SNEDDS increased Cmax and AUC, with AUC being 1.7 times higher and Cmax 2.3 times higher than the pure drug.
Zaleplon [95]	DoE: mixture design Runs: 18 Factors: % of oil, surf, and co-S Responses: globule size, % drug load	GS ↑: Oil ↑, Surf and Co-S ↓ DL ↑: Oil ↑, Surf ↓	Oil: Lavender oil (13%) Surfactant: Sorbeth-20 (49%) Co-S: HCO-60 (38%) Characteristics: GS of 87 nm, DL: 185 mg/mL	- SNEDDS showed 17 times higher drug release compared to the marketed tablet. - SNEDDS enhanced Cmax and AUC, with a relative bioavailability of 163%.
Thymoquinone [96]	DoE: mixture design Runs: 22 Factors: % of oil, surf, and co-S Responses: globule size	GS ↑: Oil ↑	Oil: Almond oil Surfactant: Tween 80 Co-S: PEG 200 Characteristics: GS 64.8 nm	- SNEDDS enhanced the pharmacological effect, with the gastroprotective index increasing approximately 2-fold.

Notes: ↑ increase in factor/response; ↓ decrease in factor/response

**Table 5 pharmaceutics-17-00701-t005:** SNEDDS optimization using response surface methodology.

API [Ref]	Experimental Design	Correlation Factors and Responses	Optimal Formulation	Product Performances
Bedaquiline [57]	DoE: Box–Benkhen design Runs: 14 Factors: % oil, Smix, sonication time Responses: globule size, PDI, % transmittance	GS ↑: Oil ↑, Surf and Co-S ↓ PDI ↑: Oil ↑, Surf and Co-S ↓	Oil: Caprylic acid (20%) Smix: Propylene glycol/Transcutol-P (40%) Sonication time: 30 S Characteristics: GS of 98.88 ± 2.10 nm, PDI of 0.3 ± 0.09, ZP of 21.16 ± 3.4 mV, ET of 15 ± 3 s	- SNEDDS showed a 93% drug release, significantly better than the suspension (55%). - SNEDDS enhanced cytotoxicity, significantly reducing the IC50 value compared to the pure drug in A549 cells.
Cinacalcet hydrochloride [58]	DoE: Box–Behnken design Runs: 15 Factors: mg of oil, surf, and co-S Responses: drug release, emulsification time, globule size, PDI	%DR ↑: Suf ↑ ET ↑: Surf ↓ GS ↑: Co-S ↑ PDI ↑: Surf and Co-S ↓	Oil: Capmul MCM (50 mg) Surfactant: Tween 20 (150 mg) Co-S: Transcutol P (55 mg) Characteristics: GS of 89.5 nm, PDI of 0.211, ET of 23.3 s	- SNEDDS showed improved drug dissolution, 2.5 times higher compared to the pure drug. - SNEDDS enhanced Cmax and AUC, with AUC increasing by 36 times compared to the pure drug.
Bosentan [47]	DoE: Box–Behnken design Runs: 15 Factors: % of oil, Surf, and co-S Responses: globule size, PDI	GS ↑: Oil ↑, Surf ↓ PDI ↑: Oil ↑, Surf ↓	Oil: Maisine 35-1 (10%) Surfactant: Kolliphor RH 40 (81%) Co-S: Labrasol (9%) Characteristics: GS of 17.11 nm, PDI of 0.180, ET < 1 min	- SNEDDS showed a 3- to 4.94 time increase in drug release compared to the reference tablet. - SNEDDS enhanced drug permeation, with a 3.36- to 16.6-time increase compared to the reference tablet. - SNEDDS improved Cmax and AUC, enhancing AUC_0–24_ and Cmax by 2.12-fold and 1.67-fold, respectively, relative to the reference.
Cephalexin [102]	DoE: Box–Behnken design Runs: 17 Factors: % of oil, surf, and co-S Responses: globule size, % transmittance, emulsification time	GS ↑: Oil ↑, Surf and Co-S ↓ ET↑: Oil ↑, surf co-S↓ %T ↑: Oil ↓, Surf and Co-S ↑	Oil: Lauroglycol 90 (14%) Surfactant: Poloxamer 188 (59%) Co-S: Transcutol-HP (32%) Characteristics: GS 87.25 3.16 nm, PDI of 0.25, ZP of 24.37 mV, ET of 52 1.7 s	- SNEDDS showed an increased drug release (94.28% vs. 25.76% for pure drug). - SNEDDS enhanced drug permeation flux (11.24 g/cm^2^/h vs. 2.84 g/cm^2^/h for pure drug). - SNEDDS exhibited improved antibacterial effects, with an increased inhibition zone. - SNEDDS enhanced Cmax and AUC, increasing bioavailability by 3.48 times compared to the pure drug.
Curcumin [103]	DoE: Box–Behnken design Runs: 17 Factors: µL of oil, surf, and co-S Responses: PDI, zeta potential, % drug loading, globule size Responses:	GS ↑: Oil ↑, Smix ↓ PDI ↑: Oil ↑, Smix ↓	Oil: Labrafil M 1994 CS (100 µL) Surfactant: Tween 80 (450 µL) Co-S: Transcutol P (450 µL) Characteristics: PDI of 0.14; ZP of −22.3 mV; DL of 95.9% and GS of 76.10 nm)	- SNEDDS increased drug dissolution by 2.22-fold compared to the pure drug. - SNEDDS enhanced drug permeation by 3.44-fold compared to the pure drug.
Ezetimibe [115]	DoE: Box Behnken design Runs: 15 Factors: % of oil, surf, and co-S Responses: globule size; %Transmittance, Emulsification time, % drug release	GS ↑: Oil ↑, Surf and Co-S ↓ ET↑: Oil ↑, surf co-S↓	Oil: Peceol (10%) Surfactant: Tween 80 (60%) Co-S: transcutol P (27%) Characteristics: GS of 24.4 ± 2.07 nm, ET of 55 s	- SNEDDS achieved 95.27% drug release in vitro.
Morin [104]	DoE: Box–Behnken design Runs: 17 Factors: Smix (ratio), % oil, Dosage Responses: Globule size, PDI	GS ↑: oil and dosage ↑, Smix ratio ↓ %DR ↑: Oil ↓, Smix ↑	Oil: GTCC (21%) Surfactant/Co-S: Cremophor RH 60/PEG 400 (1:5) Characteristics: GS of 32.8 nm, PDI of 0.089	- SNEDDS enhanced Cmax and AUC, increasing them by 10.43 times compared to the suspension. - SNEDDS improved pharmacological effects, including enhanced liver targeting, better alcohol metabolism, and greater protection against alcohol-induced gastric damage compared to the pure drug
*Passiflora ligularis* leaves extract [105]	DoE: Box–Behnken design Runs: 27 runs Factors: oil, surf, co-S, polymer Responses: globule size, PDI, Z-potential	GS↑: Oil, co-S, polymer ↑, Surf ↓ ZP ↑: Surf ↑	Oil: Castor oil (31) Surfactant: Cremophor EL (120) Co-S: Propylene glycol (80) Polymer: 30 Characteristics: GS of 22.371 ± 0.387, PDI 0.27 ± 0.03, and ZP −10.92 ± 0.42 mV	- SNEDDS exhibited enhanced pharmacological activity, showing better hypoglycemic effects than pure extract.
Piperine [106]	DoE: Box–Behnken design Runs: 17 Factors: % of oil, surf, and co-S Responses: globule size, % transmittance, emulsification time	GS ↑: Oil ↑, Surf and Co-S ↓ ET ↑: Oil ↑, Surf and Co-S %T ↑: Oil ↓, Surf and Co-S ↑	Oil: Glyceryl monolinoleat (25%) Surfactant: Poloxamer 188 (46%) Co-S: Transcutol HP (25%) Characteristics: GS of 70.34 ± 3.27 nm, ET of 53 ± 2 s	- SNEDDS showed a significant increase in drug release (97.87% vs. 27.87% compared to the pure drug). - SNEDDS demonstrated enhanced pharmacological effects, with superior antimicrobial and antioxidant activity compared to the pure drug. - SNEDDS improved Cmax and AUC, resulting in a 4.92-fold higher relative bioavailability than the pure drug.
Vemurafenib [9]	DoE: Box–Behnken design Runs: 15 Factors: % oil, surf, co-S Responses: globule size, PDI, %transmittance	GS ↑: Oil, Surf, co-S ↓ PDI ↑: Oil and Surf ↑, co-S ↓	Oil: Capryol 90 (16%) Surfactant: Tween 80 (65%) Co-S: Transcutol HP (25%) Characteristics: GS of 43.27 ± 5 nm., PDI of 0.276 ± 0.005, ZP of −8.2 mV	- SNEDDS achieved 80% drug release within the first 5 min, significantly higher than the pure drug (<20%). - SNEDDS improved Cmax and AUC, resulting in a 2.13-fold increase in relative bioavailability compared to the pure drug.
Benidipine [38]	DoE: Central Composite Design Runs: 15 Factors: % oil, surf, co-S Responses: emulsification time, globule size, % drug release, % transmittance	NA	Oil: Labrafil M 2125 CS (30%) Surfactant: Kolliphor EL (38%) Co-S: Transcutol P (40%) Characteristic: GS of 156.20 ± 2.40 nm, PDI of 0.25, ZP of −17.36 ± 0.18 mV, ET of 65.21 ± 1.95 s	- SNEDDS achieved a drug release of 92.65 ± 1.70% within 15 min, significantly higher than the marketed product (40–45%). - SNEDDS demonstrated better reductions in systolic and diastolic blood pressure compared to the control.
Docetaxel [59]	DoE: Central composite rotatable design Runs: 17 Factors: vol oil, vol Smix, sonication time (s) Responses: globule size, PDI, %transmittance, emulsification time	S ↑: Oil ↑, Smix ↓, ST ↓ PDI ↑: Oil ↑, Smix ↓, ST ↓ %ET ↑: Oil ↑, Smix ↓, ST ↓ %T ↑: Oil ↓, Smix ↑	Oil: Fish oil (0.145 mL) Smix: Tween 80/PEG 400 (0.92 mL) Sonication time: 15 s Characteristic: GS of 121.5 nm, PDI of 0.338, and ET of 22 s	- SNEDDS showed a 2-fold increase in drug release compared to the suspension. - SNEDDS demonstrated a 1.68-fold increase in drug permeation compared to the suspension.
Plumbagin [107]	DoE: Central Composite Design Runs: 14 Factors: %Oil, Smix ratio Responses: globule size, emulsification time, %drug release, equilibrium solubility	GS ↑: Oil ↑, Smix ↓ %ET ↑: Oil ↑, Smix ↓ %DR ↑: Oil ↓, Smix ↑	Oil: Capmul MCM Smix (Tween 20/PPG) ratio: 1.35:1 Characteristics: GS of 58.500 ± 1.170 nm, PDI of 0.228 ± 0.012, ET of 17.660 ± 1.520 s, ZP of −28.20 ± 1.20 mV	- SNEDDS exhibited a 93.48% drug release. - SNEDDS showed a 90.36 ± 2.78% drug permeation, significantly higher than the suspension (46.58 ± 2.10%). - SNEDDS enhanced the inhibition of oedema by 67%, compared to 36% with the suspension. - SNEDDS increased Cmax and AUC by 5 times compared to the pure drug.
Quetiapine Fumarate [108]	DoE: Central Composite Design Runs: 22 Factors: %Oil, Smix ratio, Surf type Responses: globule size, emulsification time	GS ↑: Oil ↑, Smix ↓ %ET ↑: Oil ↑, Smix ↓	Oil: Capmul MCM (10%) Smix (Gelucire 48/16/PPG) ratio: 4:1 Characteristics: GS of 92.27 nm, ET of 3.38 min, PDI of 0.32 ± 0.02, ZP of −17.1 ± 3.4 mV	- SNEDDS exhibited an extended release profile for 24 h.
Stiripentol [109]	DoE: Central Composite Design Runs: 13 Factors: % Oil, Smix ratio Responses: globule size, zeta potential, % drug release	GS ↑: oil ↑, Ratio Smix ↓ ZP ↑: oil ↓, Ratio Smix ↑ %DR ↑: oil ↑, Ratio Smix ↓	Oil: ethyl oleate (40%) Surfactant: Cremophor RH 40 (43%) Co-S: 1,2-propanediol (17%) Characteristics: GS of 45.52 ± 1.99 nm, ZP of −21.67 ± 0.24 mV, and PDI of 0.076 ± 0.011	- SNEDDS showed increased drug release (93.6 ± 2.1% vs. 38.7 ± 2.1% at 1 h) compared to the suspension. - SNEDDS enhanced Cmax and AUC (relative BA 218.01% compared to suspension).
Tamoxifen and Resveratrol [62]	DoE: Central composite rotatable design Runs: 13 Factors: vol of oil and Smix Responsed: globule size; PDI, %transmittance	GS ↑: Cons of Oil ↑, Smix ↓ PDI ↑: Smix ↓ %T ↑: Oil ↓, Smix ↑, ST ↑	Oil: Capmul MCM (0.6 mL) Smix: Tween 80/Transcutol-HP (1.86 mL) Characteristics: GS of 104.5 nm and PDI of 0.211	- SNEDDS showed a significantly higher drug release (85% after 720 min) compared to the suspension. - SNEDDS enhanced drug permeation (2.3-fold higher) and improved therapeutic efficacy and internalization in the MCF-7 cell line compared to the suspension. - SNEDDS increased Cmax and AUC (1.66 and 1.63 times higher, respectively) compared to the suspension.
Venetoclax [110]	DoE: Central Composite Design Runs: 32 Factors: mg of oil, surf, co-surf, rate of stirring, stirring time Responses: globule size, PDI, emulsification time, % transmittance	GS ↑: Oil ↑ PDI ↑: Oil ↑, Surf ↓ ET: Oil ↑, Smix ↓ %T ↑: Surf ↑	Oil: Cinnamon oil (100 mg) Surfactant: Cremophor RH40 (300 mg) Co-S: Transcutol P (250 mg) Rate of stirring: 150 rpm Stirring time: 10 min Characteristics: GS of 71.32 ± 2.85 nm, PDI of 0.113 ± 0.01, ET of 16.4 ± 0.81 s	- SNEDDS achieved a 96.91 ± 3.8% drug release within 2 h, compared to only a 13.87 ± 0.84% release from the pure drug. - SNEDDS showed 3–7-fold lower IC₅₀ and enhanced anticancer activity compared to pure drug, with greater cellular uptake, apoptosis induction, and modulation of apoptotic markers. - SNEDDS increased Cmax by ~5-fold and enhanced oral bioavailability over the suspension in vivo.
Olmesartan Medoxomil [60]	DoE: D-optimal response surface design Runs: 16 Factors: mg of oil, surf, and co-surf Responses: globule size, emulsification time, % drug release	GS ↑: Oil and Co-S ↑, Surf ↓ %DR ↑: Cons of Oil Surf ↓	Oil: Capmul MCM (393 mg) Surfactant: Tween 80 (423 mg) Co-S: Transcutol HP (184 mg) Characteristics: GS of 64.2 nm and ZP of −25.4 mV	- SNEDDS achieved a >85% release in 30 min, higher than the pure drug (~43%). - SNEDDS increases 5.06–5.58-fold in Cmax and 3.87–4.21-fold in AUC and a reduction in Tmax, compared to the pure drug.

Notes: ↑ increase in factor/response; ↓ decrease in factor/response.

**Table 6 pharmaceutics-17-00701-t006:** SNEDDS optimization using factorial design.

API [Ref]	Experimental Design	Correlation Factors and Responses	Optimal Formulation	Product Performances
Camptothecin [116]	$\mathrm{DoE}:\;3^{2}$ Full Factorial Design Runs: 13 Factors: % of oil, Smix Responsed: % transmittance, emulsification time, % drug release	%T ↑: Oil ↓, Smix ↑: ET ↑: Oil ↑, Smix ↓ %DR ↑: Oil ↓, Smix ↑,	Oil: Omega oil (12.5%) Smix: Cremophor RH40/Labrafil M2125 (40%) Characteristics: GS of 47 nm, PDI of 0.176, ZP of 35.2 mV, and ET of 11 s	- SNEDDS demonstrated a significantly higher drug release (99.33 ± 0.26% within 20 min) compared to other formulations and pure drug (22.72%). - SNEDDS significantly (*p* < 0.05) inhibited BEWO cell proliferation more effectively than the pure drug. - SNEDDS significantly enhanced the oral bioavailability 17-fold compared to the pure drug.
Exendin-4 [117]	$\mathrm{DoE}:\;3^{2}$ factorial D-optimal design Runs: 13 Factors: gram of oil, Smix Responses: globule size, PDI, Z-potential	DS ↑: Oil ↑, Surf ↓ PDI ↑: Oil ↑, Surf ↓ ZP ↑: Oil ↑, Surf ↓	Oil: Ethyl oleate (15.42) Surfactant: Cremophor EL (42.5) Co-S: Labrasol/ethanol (21/5) Characteristics: GS of 20.66 ± 0.40 nm, PDI of 0.1 ± 0.01 and ZP of −5.99 ± 0.43 mV	- SNEDDS increased the permeability coefficient by 1.5-fold compared to the plain solution.
Flufenamic Acid [118]	DoE: 3^2^ Full Factorial Design Runs: 9 Factors: mg of oil, Surf Responses: globule size, Z-potential, PDI	GS ↑: Oil ↑, Surf ↓	Oil: Miglyol 812 (150 mg) Smix: Labrasol/Cremophor EL (300 mg) Characteristics: GS of 61.12 nm, ZP of −25.53 mV and PDI of 0.432	- SNEDDS showed a higher drug release (91.38 ± 5.34%) than FLF suspension (11.38 ± 1.12%) within 2 h. - SNEDDS showed a significantly higher anti-inflammatory activity than pure FLF, with % inhibition of 25.68 ± 2.68% vs. 6.42 ± 0.78% at 1 h.
Fosfestrol [119]	$\mathrm{DoE}:\;3^{2}$ Full Factorial Design Runs: 13 Factors: % of oil and Smix Responses: % transmittance, emulsification time, % drug release	%T ↑: Oil ↓, Smix ↑: ET ↑: Oil ↑, Smix ↓ %DR ↑: Oil ↓, Smix ↑	Oil: Soyabean oil (10%) Smix: Labrasol ALF/Labrafil-M212 (39.48%) Characteristics: GS of 52 nm, PDI of 0.158, ET 24 s	- SNEDDS showed a significantly higher drug release (98.20 ± 1.3%) compared to pure drug (32.0 ± 3.3%) and the marketed formulation (40.36 ± 2.8%). - SNEDDS exhibited a 4.68-fold increase in Caco-2 monolayer permeability (62.8 × 10⁻⁶ cm/s) compared to pure drug (13.4 × 10⁻⁶ cm/s). - SNEDDS demonstrated significantly higher cytotoxicity, apoptosis induction, mitochondrial membrane potential disruption, and G2/M phase cell cycle arrest. - SNEDDS demonstrated a 4.5-fold increase in bioavailability compared to pure drug.
Isoliquiritigenin [120]	DoE: 2^2^ Full Factorial Design Runs: 9 Factors: % oil, Smix ratio Responses: globule size, % drug loading	%GL ↑: Oil ↑, Ratio Smix ↑	Oil: Ethyl oleate Surfactant: Tween 80 Co-S: PEG 400 Characteristics: GS of 20.63 ± 1.95 nm, PDI of 0.11 ± 0.03, and ZP of −12.64 ± 2.12 mV	- SNEDDSS exhibited a significantly higher cumulative release (70.13 ± 3.47%) than ILQ suspension (37.25 ± 4.09%) after 24 h. - SNEDDS significantly enhanced intestinal absorption, with a 1.6-fold increase compared to suspension. - SNEDDS resulted in a 3.92-fold increase in bioavailability compared to suspension. - SNEDDS demonstrated better anti-asthma efficacy compared to suspension.
Lamotrigine [125]	DoE: D-optimal factorial design Runs: 19 Factors: % oil, co-S type, Smix ratio Responses: globule size, % drug release	NA	Oil: Rose oil (30%) Surfactant: Cremophor EL (47%) Co-S: PEG 400 (23%) Characteristics: GS of 15.013 ± 0.158 nm, PDI of 0.245 ± 0.018, ZP of −7.97 mV	- SNEDDS significantly enhanced drug release to 92.7%, compared to only 7.88% from the pure drug. - SNEDDS increased the bioavailability 2.03-fold compared to pure drug and 1.605-fold compared to marketed tablet.
Palbociclib-letrozole [121]	$\mathrm{DoE}:\;3^{2}$ Full factorial Design Runs: 9 Factors: %oil, Smix Responses: % transmittance, emulsification time, % drug release	%T: Oil ↓, Smix ↑: ET ↑: Oil ↑, Smix ↓ %DR ↑: Oil ↓, Smix ↑	Oil: Maisine (10%) Smix: Cremophor-RH40/Labrasol (35%) Characteristics: GS of 71 ± 3 nm, PDI of 0.28 ± 0.09, ET of 80 ± 1.13 s, ZP: 31 ± 2 mV	- SNEDDS significantly enhanced the drug release of PCB (99.76 ± 0.75%) and LTZ (94.86 ± 2.37%) within 20 min compared to plain dispersions (20.14 ± 1.52% and 31.45 ± 1.03%). - SNEDDS enhanced the intestinal permeability of PCB and LTZ, as shown by a ~4-fold and ~1.7-fold increase in Papp values. - SNEDDS demonstrated superior cytotoxicity and apoptosis in breast cancer cell lines. - SNEDDS significantly enhanced the oral bioavailability approximately 7.5-fold and 4-fold.
Rhein [122]	$\mathrm{DoE}:\;3^{2}$ Full Factorial Design Runs: 10 Factors: % oil, Smix Responses: globule size, % transmittance, emulsification time	GS ↑: Oil ↑, Smix ↓ %T ↑: Oil ↓,Smix ↑: ET ↑: Oil ↑, Smix ↓	Oil: Eucalyptus oil (50%) Smix: Tween 80/PEG 400 (50%) Characteristics: GS of 129.3 ± 1.57 nm, ZP of −24.6 mV ± 0.34, EE of 98.86 ± 0.23%.	- SNEDDS enhanced drug release compared to pure drug (98.18 ± 1.04% vs. 23%). - SNEDDS significantly enhanced the bioavailability of rhein, with Cmax and AUC values in SNEDDS being approximately 4.1 times and 5.2 times higher compared to pure drug.
Sertraline [123]	$\mathrm{DoE}:\;2^{3}$ Full Factorial Design Runs: 8 Factors: mg of oil, surf, Co-S Responses: dissolution efficiency, globule size, and emulsification time	DE ↑: oil and co-S ↓, Surf ↑ GS ↑: surf ↑ ET ↑: oil ↑, Smix ↓	Oil: Glycerol triacetate (100 mg) Surfactant: Tween 80 (133 mg) Co-S: PEG 200 (66 mg) Characteristics: GS of 76.03 nm, ET of 29 s, PDI of 0.422, and ZP of 25.5 mV	- SNEDDS enhanced the release with 82.76% of drug release in 30 min compared to only 12.56% for pure drug. - SNEDDS enhanced Cmax and AUC, resulting in a 386% increase in relative bioavailability compared to suspension.
Zolmitriptan [124]	$\mathrm{DoE}:\;3^{2}$ Full Factorial Design Runs: 9 Factors: % oil and Smix Responses: globule size, zetapotential, % drug release	GS ↑: Oil ↑, Surf ↓ ns of Oil ↑, Surf ↓	Oil: Lavender oil (30%) Smix: P35HC/Transcutol HP (20%) Characteristics: GS of 19.59 ± 0.17 nm, ZP −23.5 ± 1.17 mV, and ET of 121 ± 2.51 s	- SNEDDS exhibited rapid drug release, being almost complete within the first 10 min. - SNEDDS significantly enhanced permeation compared to the solution. - SNEDDS improved effects on the psychological state, algesia, and maintained normal brain electrical activity.

Notes: ↑ increase in factor/response; ↓ decrease in factor/response.

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
