# Peer review of "Quality by Design and In Silico Approach in SNEDDS Development: A Comprehensive Formulation Framework"

_pharmaceutics, 2025, doi:10.3390/pharmaceutics17060701_

Round 1

Reviewer 1 Report

Comments and Suggestions for Authors

This is an excellent review article, linking the formulation of SNEDDS with Quality by design in a comprehensive manner. The review article would be of interest to the readers of Pharmaceutics. It addresses a knowledge gap of lack of review articles describing development of SNEDDS, in a way which facilitates their large scale production. References are adequate and relevant to the topic. The following minor comments need to be addressed before the manuscript can be accepted for publication:

1- Table 4 please correct the spelling of glimepirid to glimepiride

2- Section 2, please explain how are SNEDDS different from microemulsions

3- In the conclusion section, please delineate whether the Quality by design approaches describes are satisfactory in overcoming the challenges mentioned by the authors in lines 53-56.

Author Response

We would like to sincerely thank the reviewer for the valuable comments and suggestions, which have significantly helped us improve the quality and clarity of the manuscript. Below, we provide a point-by-point response to the comments. We hope that our revisions are satisfactory.

Comment 1: Table 4, please correct the spelling of glimepirid to glimepiride

Response 1: Thank you for pointing this out. Therefore, we have corrected the spelling in Table 4 from 'glimepirid' to 'glimepiride' as suggested.

Comment 2: Section 2, please explain how are SNEDDS different from microemulsions

Response 2: Thank you for pointing this out. We have added an explanation in Section 2 regarding the differences between SNEDDS and nanoemulsion/microemulsion systems. 

Comment 3: In the conclusion section, please delineate whether the quality by design approaches described are satisfactory in overcoming the challenges mentioned by the authors in lines 53-56.

Response 3: Thank you for pointing this out. We have updated the conclusion to clarify if the QbD approaches mentioned in the manuscript successfully tackle the formulation issues, especially the complicated relationships between SNEDDS components and how they affect stability, emulsification efficiency, and drug loading. The revised conclusion now explicitly reflects how QbD strategies contribute to addressing these challenges.

Reviewer 2 Report

Comments and Suggestions for Authors

The authors provide an in-depth and structured discussion of theoretical principles, practical formulation strategies, and critical analysis of recent studies (2020–2025) incorporating QbD and DoE frameworks.

The topic is highly relevant and timely, especially considering the increasing importance of robust, science-based formulation development in pharmaceutical sciences. The review effectively compiles and synthesizes a broad range of literature and proposes a methodological framework for SNEDDS development under QbD principles.

  1. While the article compiles valuable information, many descriptions (e.g., definitions of SNEDDS, functions of oil/surfactant/co-surfactant) repeat content already well established in prior reviews. Focus more on comparative or critical analysis rather than descriptive repetition, especially in early sections.
  2. Writing at times lacks academic refinement and includes minor grammatical errors.

Example: “DoE has become familiar with being applied…” should be revised to “DoE has become widely applied…”.

  1. Tables 4 and 5 are detailed and informative but extremely lengthy. Consider moving some of the more extensive data tables to supplementary material and summarizing key trends in the main text.
  2. The manuscript presents numerous studies but lacks critique of study design quality, reproducibility, or limitations of QbD application in real-world SNEDDS development. Include a subsection discussing gaps in current QbD/DoE practices, limitations of cited studies, or challenges in industrial translation.
  3. Although the review is forward-looking, it does not sufficiently address recent trends in AI/ML-based optimization, regulatory harmonization efforts, or potential in pediatric/geriatric formulations. Expand the conclusion or add a "Future Outlook" section addressing such opportunities.

Author Response

We would like to sincerely thank the reviewer for the valuable comments and suggestions, which have significantly helped us improve the quality and clarity of the manuscript. Below, we provide a point-by-point response to the comments. We hope that our revisions are satisfactory.

Comment 1: While the article compiles valuable information, many descriptions (e.g., definitions of SNEDDS, functions of oil/surfactant/co-surfactant) repeat content already well established in prior reviews. Focus more on comparative or critical analysis rather than descriptive repetition, especially in early sections.

Response 1: Thank you for pointing this out. We have revised Section 2 to make the theoretical background of SNEDDS more concise.

Comment 2: Writing at times lacks academic refinement and includes minor grammatical errors

Response 2: Thank you for your valuable feedback. We have thoroughly revised the manuscript to enhance its academic tone and clarity of language. Minor grammatical errors have been corrected. We also utilized grammar-checking tools to improve the overall readability and professionalism of the text.

Comment 3: Tables 4 and 5 are detailed and informative, but extremely lengthy. Consider moving some of the more extensive data tables to supplementary material and summarizing key trends in the main text.

Response 3: Thank you for your thoughtful suggestion. We acknowledge that Tables 4 and 5 are extensive; however, we believe that retaining them in the main manuscript is essential, as the data presented serve as a critical foundation for the subsequent discussion and comparative analysis. Nevertheless, we have carefully reviewed the layout and formatting to enhance readability and ensure that the tables remain clear and concise while supporting the main narrative of the review. We hope this decision is understandable; however, if deemed necessary, we are willing to relocate the tables to the supplementary section in the next revision.

Comment 4: The manuscript presents numerous studies but lacks a critique of study design quality, reproducibility, or limitations of QbD application in real-world SNEDDS development. Include a subsection discussing gaps in current QbD/DoE practices, limitations of cited studies, or challenges in industrial translation.

Response 4: Thank you for this insightful comment. We have included a dedicated section entitled “Limitations and Challenges” to provide a more critical perspective on the cited studies. This section discusses important issues like differences in the quality of experimental designs, not enough focus on critical process parameters (CPPs), inconsistent methods used, and the lack of testing for formulations when increasing scale and checking long-term stability.

Comment 5: Although the review is forward-looking, it does not sufficiently address recent trends in AI/ML-based optimization, regulatory harmonization efforts, or potential in pediatric/geriatric formulations. Expand the conclusion or add a "Future Outlook" section addressing such opportunities.

Response 5: Thank you for this insightful comment. We have strengthened the forward-looking perspective by expanding the "Future Perspectives" section to include additional considerations on the regulatory landscape, as well as specific formulation opportunities for pediatric and geriatric populations. These additions are intended to reflect the evolving translational needs and regulatory expectations for SNEDDS, particularly in vulnerable patient groups. We have also incorporated a discussion on in silico approaches, machine learning (ML), and artificial intelligence (AI) as emerging tools to support formulation optimization and prediction. (The section was located after the conclusion.)

Reviewer 3 Report

Comments and Suggestions for Authors

Thank you for this opportunity to review. This review article titled " Quality by Design in SNEDDS Development: A Comprehensive Formulation Framework" could contribute significant updates on the topic of SNEDDS formulation development. The overall structure of the article is well-defined, however, there needs some revisions especially before consideration for its acceptance. The major comments are included below:

  1. Line 163-165: what is the author's comment on the type of aqueous medium used to mix with the preconcentrate. Should it be purely water or something that is more biorelevant such as PBS? What about the pH of the medium? Since most cases SNEDDS are taken orally into the GIT.
  2. Table 2: A suggestion to add lines between each row of the table. Currently, the justification column is all bunched together for each QTPP element, making it hard to read.
  3. How about considerations for SNEDDS that are not orally administered? would the general QTPP, CMAs, CPPs, and CQAs be different?
  4. Line 309 typo: "QBD" should be "QbD."
  5. What are some considerations about the safety of the excipients used in SNEDDS formulations? Are there certain CMAs that are being followed to ensure safety of certain excipients?
  6. What about considerations for scale-up process for SNEDDS manufacturing? How would QbD be different in terms of QTPP, CMAs, CPPs, and CQAs?
  7. Using the QbD approach for formulation optimization is not a new topic for SNEDDS. I would suggest the authors to increase bulk of the review article on more novel technologies used for SNEDDS development, such as the few examples that were mentioned in "future perspectives." As AI is a very hot topic in formulation sciences presently, emphasizing on such topics would draw more attention and readership.

Author Response

We would like to sincerely thank the reviewer for the valuable comments and suggestions, which have significantly helped us improve the quality and clarity of the manuscript. Below, we provide a point-by-point response to the comments. We hope that our revisions are satisfactory.

Comment 1: Line 163-165: what is the author's comment on the type of aqueous medium used to mix with the preconcentrate. Should it be purely water or something that is more biorelevant such as PBS? What about the pH of the medium? Since most cases SNEDDS are taken orally into the GIT.

Response 1: Thank you for pointing this out. In many SNEDDS studies discussed in this article, the emulsification efficiency test, which is used to help choose the right surfactant and co-surfactant combinations, is usually done with distilled water. We have added references in the revised manuscript to support this practice. However, we agree that using biorelevant media is crucial for more accurate prediction of in vivo performance. Usually, after finding the best formulation, we test how well it holds up when diluted by using media that better mimic the body's environment, like 0.1 N HCl and phosphate buffer at pH 6.8, which represent the conditions in the stomach and intestines.

Comment 2: Table 2, A suggestion to add lines between each row of the table. Currently, the justification column is all bunched together for each QTPP element, making it hard to read.

Response 2: Thank you for pointing this out. We have revised Table 2 by adding horizontal lines between each row to improve readability and ensure more precise separation of the justification column content for each QTPP element.

Comment 3: How about considerations for SNEDDS that are not orally administered? would the general QTPP, CMAs, CPPs, and CQAs be different?

Response 3: Thank you for pointing this out. Most studies develop SNEDDS primarily for oral administration. However, a few studies are exploring their application via alternative routes such as ocular, intranasal, or transdermal delivery, although these remain relatively limited in number. For these non-oral routes, some differences in the QTPP, CMAS, CPPS, and CQAS may be considered. While some formulation parameters may overlap with oral SNEDDS, the target product profile and performance requirements are highly route-specific. We have included a clarification in section 4.3, to explain this point, along with an example of SNEDDS formulation for eye use using DoE, to show how the way a product is given requires special attention in its formulation and assessment.

Comment 4: Line 309 typo: "QBD" should be "QbD."

Response 4: Thank you for pointing this out. We have corrected the typo from "QBD" to "QbD"

Comment 5: What are some considerations about the safety of the excipients used in SNEDDS formulations? Are there certain CMAs that are being followed to ensure the safety of certain excipients?

Response 5: Thank you for pointing the issue out. We have added a discussion regarding the safety of excipients in SNEDDS formulations in the revised manuscript in Section 2 (Basic Principles of SNEDDS Formulation). Safety is indeed a critical consideration, especially for oral administration, where excipients are often used at relatively high concentrations. We highlighted the importance of selecting excipients with GRAS status or those listed in the FDA Inactive Ingredient Database for oral use. Under the QbD approach, this safety consideration is also integrated into the design space and reflected in the choice of CMA.

Comment 6: What about considerations for the scale-up process for SNEDDS manufacturing? How would QbD be different in terms of QTPP, CMAs, CPPs, and CQAs?

Response 6: Thank you for pointing that out. We have added a discussion on scale-up considerations for SNEDDS manufacturing in section 9 (challenge and limitation) and section 11 (future perspective).  We explain that while QTPPs and CQAs generally remain consistent during scale-up, the most significant adjustments typically occur in CPPs to ensure process robustness and product quality at larger scales. CMAs may also require minor refinements due to variability in raw materials. This approach aligns with the QbD framework to maintain product performance and patient safety throughout the manufacturing scale-up process.

Comment 7: Using the QbD approach for formulation optimization is not a new topic for SNEDDS. I would suggest the authors to increase bulk of the review article on more novel technologies used for SNEDDS development, such as the few examples that were mentioned in "future perspectives." As AI is a very hot topic in formulation sciences presently, emphasizing on such topics would draw more attention and readership.

Response 7: Thank you for pointing that out. We appreciate your suggestion regarding the inclusion of more novel technologies in SNEDDS development. As mentioned in the Future Perspectives section, we have discussed how important silico modeling, especially molecular modeling, is becoming and how it can work with machine learning to help create better formulations. We have revised the title, abstract, and introduction to reflect the emphasis on these advanced technologies better. Additionally, we have included a section elaborating on the role of in silico tools and molecular modeling in SNEDDS development. As for the application of AI-driven formulation platforms (e.g., FormulationAI), we acknowledge that current literature on their direct implementation in SNEDDS is limited. Therefore, we have retained this topic in the Future Perspectives section as a promising direction for future research.
